# Off-Policy Evaluation with Strategic Agents via Local Disclosure

**Kiet Q. H. Vo** [1]   **Abbavaram Gowtham Reddy** [1]   **Julian Rodemann** [1 2]   **Siu Lun Chau** [3]   **Krikamol Muandet** [1]

## Abstract

We study off-policy evaluation (OPE) under strategic behavior where decision subjects (or agents) respond to a decision maker's policy by strategically modifying their covariates. Such behavior induces a *policy-dependent* covariate shift, breaking the standard assumption in existing methods that covariates are exogenous to the policy. Related work addresses this challenge by imposing strong assumptions such as repeated interactions or full knowledge of agents' response behavior, substantially limiting its applicability to OPE. In contrast, we consider a one-shot OPE setting where the decision maker has only partial knowledge of the agents' response behavior. Our key insight is that disclosing local information through post-hoc explanations reveals agents' pre-strategic covariates prior to adaptation, mitigating the information loss induced by strategic behavior. Leveraging this structure, we estimate a statistical model for the agents' responses and construct a doubly robust estimator for policy value. By assuming that the agents' cost sensitivity follows a conditional log-normal distribution, we establish consistency of the proposed estimator and validate our approach empirically. More broadly, our results highlight how interaction design can mitigate information asymmetry by revealing otherwise hidden structure in agents' strategic responses.

## 1. Introduction

The abundance of individual-level data has made it increasingly feasible for decision makers (DMs) to design and deploy personalized policies across a wide range of domains including healthcare (Murphy, 2003; Hamburg & Collins,

2010), education (Mandel et al., 2014), lending (Kilbertus et al., 2020), and recommendation systems (Joachims et al., 2021). In these applications, particularly in high-stakes settings such as healthcare, deploying new policies directly is generally prohibitive, as poorly chosen decisions may lead to substantial harm, financial loss, or unfair outcomes. As a result, rather than evaluating through experimentation, policy performance must be assessed using historically collected data generated under a different decision policy. This problem is commonly referred to as an *off-policy evaluation* (OPE) problem (Uehara et al., 2022).

When decisions are personalized, agents, i.e., individuals subject to the decision rule, may strategically modify their observable covariates to receive more favorable decisions. For instance, if college admissions policies place greater weight on standardized test scores (e.g., GRE), applicants may reallocate effort toward those tests (Vo et al., 2024); similarly, when a bank introduces a new lending policy, customers might alter their financial behavior to meet eligibility criteria (Tsirtsis & Gomez Rodriguez, 2020). Such strategic responses induce a *policy-dependent* shift in the distribution of agents' covariates: as the policy changes, so does the population it acts upon. This phenomenon is widely studied in strategic classification (Hardt et al., 2016) and performative prediction (Perdomo et al., 2020). Because policy performance is typically defined as an average over the induced population, a central insight from this literature is that evaluating a new policy requires *anticipating how the covariate distribution responds to it*; failing to do so amounts to evaluating the policy under an incorrect induced population and thus estimating the wrong policy value. Nonetheless, this challenge is largely overlooked in OPE literature.

Prior work on strategic behavior in offline settings primarily originates from the strategic classification literature (Hardt et al., 2016; Levanon & Rosenfeld, 2021; Rosenfeld & Rosenfeld, 2024). To enable tractable analysis and equilibrium characterization, these works typically assume that agents' responses can be modeled precisely or that their behavior is homogeneous, commonly formalized through a *single* and/or *known* cost function governing covariate modification. While analytically convenient, such assumptions rarely hold in practice. Agents' preferences and constraints are inherently heterogeneous, and are generally unobserved by the DM. Hence, applying these approaches as-is to OPE

[1]CISPA Helmholtz Center for Information Security, Saarbrücken, Germany [2]Department of Statistics, LMU Munich, Germany [3]College of Computing and Data Science (CCDS), Nanyang Technological University, Singapore. Correspondence to: Kiet Q. H. Vo <huynh.vo@cispa.de>.

*Proceedings of the 43rd International Conference on Machine Learning*, Seoul, South Korea. PMLR 306, 2026. Copyright 2026 by the author(s).

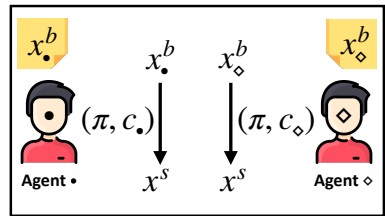
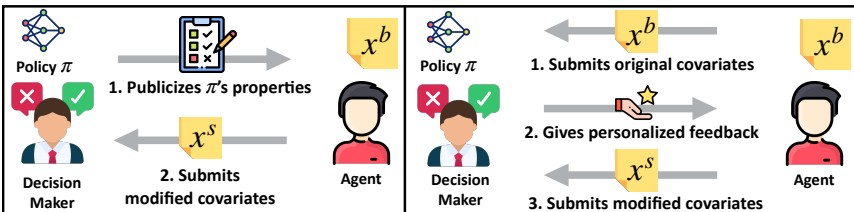

*(a)* Two agents modify to the same covariate.   *(b)* Global information disclosure (GID) vs. local information disclosure (LID).

*Figure 1.* The left figure *(a)* shows a situation where two agents—with two different pre-strategic covariates $x_\bullet^b, x_\diamond^b$ and cost functions $c_\bullet, c_\diamond$—adapt to the same covariate value $x^s$. If the pre-strategic covariates are not observed, which happens in global information disclosure (i.e., right figure *b*), these two agents are indistinguishable. This makes it harder for the DM to reason about the effect of another policy $\pi'$ on the covariate shift. The right figure *(b)* illustrates the interactions in GID (where the DM makes their policy's properties public) and LID (where the DM withholds disclosure until the agent gives information about themselves).

can substantially limit their practical usefulness, as OPE is most often employed to support real-world decision making.

Motivated by this, we study OPE under strategic behavior while relaxing the assumption of a single and known cost function. Specifically, we allow agents to have heterogeneous cost functions, while assuming that the DM knows only their common components and not the agent-specific costs. This relaxation is possible when the DM employs local information disclosure (LID): disclosing partial information about their policy as a form of personalized feedback, for instance, through post-hoc explanations (Tsirtsis & Gomez Rodriguez, 2020; Xie & Zhang, 2024; Vo et al., 2026); see Figure 1b.

Crucially, this interaction scheme allows the DM to observe agents' original covariates (or *pre-strategic covariates*) before their strategic adaptation. Observing pre-strategic covariates is essential because different agents—with different baseline characteristics and behavioral models—may strategically modify to the same final covariate value. Without access to pre-strategic information, such trajectories are observationally indistinguishable, making it impossible to separate strategic modification from genuine baseline characteristics. Figure 1a illustrates this. By contrast, under global information disclosure (GID), where the DM makes policy information public and observes only post-adaptation covariates, they lose access to pre-strategic information, e.g., as in Shavit et al. (2020); Harris et al. (2022b); Munro (2025); Cohen et al. (2024). While prior work has studied strategic behavior under LID (Tsirtsis & Gomez Rodriguez, 2020; Xie & Zhang, 2024; Vo et al., 2026), they mainly focus on online learning and equilibrium analysis. To our knowledge, no existing work leverages pre-strategic information for OPE with strategic agents, nor for one-shot learning with heterogeneous and unknown agents' behavior.

More broadly, this limitation of GID reflects a form of *information asymmetry*: the DM observes agents only after strategic adaptation and therefore lacks access to key pre-adaptation information. Although LID is used as part of our problem setup, we emphasize that, practically, LID should be understood as a design choice in how the DM structures interactions with agents, rather than as a restrictive evaluation setup introduced solely to enable estimation. In many systems, the DM has discretion over whether policy information is revealed globally or through personalized feedback, and this choice fundamentally shapes agents' strategic responses and what can be inferred from data.

Our work introduces this connection between interaction design and inferential structure as a novel perspective on OPE with heterogeneous and partially unknown agents' behavior. From this perspective, our main research question is therefore twofold: (i) under local information disclosure, how should the disclosure rule be designed for OPE under strategic behavior, and (ii) how can the corresponding policy value be estimated from historical data?

We summarize below our contributions:

- As the first work to apply LID to OPE under strategic behavior, we extend the action recommendation-based explanation (ARex) (Vo et al., 2026) and adapt its explanation rule to handle covariate shift in OPE (Lemma 2.2).
- We show that ARexes, as an instantiation of LID, can be used to infer the behavioral model of agents. Under some structural assumptions, we show that the estimator of unknown parameters is consistent (Theorem 2.6).
- We propose a doubly robust estimator that can adjust for the strategic covariate shift, and prove the estimator's consistency under standard conditions (Theorem 3.2).

All proofs are provided in the appendices.

## 2. Strategic OPE under Local Disclosure

### 2.1. Problem Formulation

We use the lending scenario (Harris et al., 2022a) as our running example and consider the setting in which a DM (e.g., a bank) interacts with a population of agents (e.g., their customers). Following prior work (Tsirtsis & Gomez Ro-

driguez, 2020; Harris et al., 2022b; Vo et al., 2024; 2026), we assume agents interact with the DM independently of one another. Therefore, for ease of exposition, we describe the setup for a single agent and simply view a population of heterogeneous agents as concrete realizations of the same model. In particular, we adopt the strategic agent setting from Vo et al. (2026) and describe it below.

Let $X^b \sim P_{X^b}$ be an agent's covariate vector and $x^b \in \mathcal{X}$ an independent realization representing the agent's observable attributes, such as existing debt or bank account balance. At this stage, as the agent has not modified their covariates, we also refer to the base covariates $x^b$ as their pre-strategic covariates. We assume the covariate space $\mathcal{X}$ is discrete. This fits many real-world scenarios where the DM dictates which agents' information is collected and where continuous values are often discretized. For example, a bank may only record coarse, pre-specified attributes such as credit score buckets, income ranges, and debt-to-income ratio categories.

In the beginning, the DM commits to a decision policy $\pi : \mathcal{X} \to [0,1]$ that controls the probability of the agent getting a positive treatment (e.g., a loan application getting approved). Let $T^b \mid x^b \sim \text{Bernoulli}(\pi(x^b))$ be a binary random variable that represents the treatment assigned to this agent, with $t^b \in \mathcal{T} = \{0,1\}$ denotes the realization. We note that our formulation of the (potentially stochastic) treatment policy $\pi$ is consistent with prior work in OPE; see, e.g., Uehara et al. (2020); Kallus et al. (2022). Moreover, this stochasticity is realistic in many situations, such as when the DM wants randomization for learning (Kilbertus et al., 2020; Munro, 2025; Vo et al., 2024) or when it is a consequence of credit rationing (Stiglitz & Weiss, 1981).

**DM's personalized feedback.** If the agent receives a negative treatment ($t^b = 0$), they are provided with personalized feedback in the form of an action recommendation-based explanation (ARex) (Vo et al., 2026) and are allowed to modify their observable features $x^b$ before reapplying. We extend the ARex framework of Vo et al. (2026) to allow the agent to receive an explanation $e$ that contains $k \geq 2$ recommendations, i.e., $e = \{(x_j^r), \pi(x_j^r)\}_{j=1}^k$, rather than being restricted to only two recommendations. In our lending example, these recommendations can inform the agent to pay off more debt or increase the amount in their savings account. This is similar to the concept of algorithmic recourse (Karimi et al., 2021; Harris et al., 2022a; König et al., 2026). In addition, we refer to this set of recommended feature updates as $\mathcal{X}^r = \{x_j^r\}_{j=1}^k$ and we use $\tau : (x^b, \pi) \mapsto e$ to denote the explanation rule that generates ARexes. For a given choice of $k$, $\mathcal{E} = \mathcal{X}^k \times [0,1]^k$ denotes the space of ARexes and $E$ denotes the (random) explanation.

**Agent's adaptation.** Following Tsirtsis & Gomez Rodriguez (2020); Vo et al. (2026), we model the agent's

strategic modification of their covariate vector as

$$x^s \in \underset{x \in \{x^b\} \cup \mathcal{X}^r}{\arg\max} \underbrace{\left\{ \pi(x) - c(x, x^b) \right\}}_{u(\pi, c, x, x^b)}, \qquad (1)$$

where $c : \mathcal{X} \times \mathcal{X} \to \mathbb{R}_{\geq 0}$ denotes the cost function of this agent for modifying their base features $x^b$ to $x$.

While this choice-based behavioral assumption might appear strong, particularly when only a single recommendation is provided, our work mitigates this concern by allowing $k \geq 2$ recommendations. Moreover, following Vo et al. (2026), we assume that agents do not explore actions outside the presented choices set $\{x^b\} \cup \mathcal{X}^r$, so as to avoid the risk of inadvertently reducing their utility. Section B.1 further discusses this behavioral assumption. We denote $\mathcal{X}^f := \{x^b\} \cup \mathcal{X}^r$ as the set of feasible actions.

**Agent's cost model.** We assume that the agent's cost function takes the form of $c(x, x') := \alpha d(x, x')$, where $d : \mathcal{X} \times \mathcal{X} \to \mathbb{R}_{\geq 0}$ is a deterministic function measuring the primitive cost of modifying covariates $x$ to $x'$. The function $d$ is shared across agents and known by the DM, while the scalar $\alpha \in (0, \infty)$ captures agent-specific cost sensitivity, unknown to the DM. To model heterogeneity across agents, we assume $\alpha$ follows a log-normal distribution:

$$\ln \alpha \mid x^b \sim \mathcal{N}(\beta^\top \phi(x^b) + \beta_0, \sigma^2), \qquad (2)$$

where $\phi : \mathcal{X} \to \mathbb{R}^p$ denotes some known feature transformation function of the covariate vector $x^b$. Furthermore, the conditional CDF $F(\alpha \mid x^b; \theta)$ is parameterized by $\theta = (\beta, \beta_0, \sigma) \in \Theta \subseteq \mathbb{R}^p \times \mathbb{R} \times \mathbb{R}^+$.

Intuitively, $d(x, x')$ captures the underlying burden of modifying covariates (e.g., time, effort, or monetary expenses), while $\alpha$ reflects how different agents internalize this burden. The log-normal model provides a tractable parameterization of heterogeneous cost sensitivities in the one-shot setting.[1] We rewrite the agent's strategic adaptation as follows:

$$x^s \in \underset{x \in \mathcal{X}^f}{\arg\max} \underbrace{\left\{ \pi(x) - \alpha d(x, x^b) \right\}}_{u(\pi, \alpha, x, x^b)}$$

**Agent's outcome.** If the agent modifies their feature vector to a new value $x^s \neq x^b$ and reports $x^s$ to the DM, they receives the final treatment $t^s \in \mathcal{T} = \{0,1\}$, modeled as $T^s \sim \text{Bernoulli}(\pi(x^s))$. Then, the agent's outcome is realized as $y := h(x^s, t^s, z)$ where $y \in \mathcal{Y} \subseteq \mathbb{R}$, $h : \mathcal{X} \times \mathcal{T} \times \mathcal{Z} \to \mathcal{Y}$ denotes the (non-random) outcome function, and $z \in \mathcal{Z}$ the unobserved noise factor. In our lending example, this outcome corresponds to the profit the bank makes from approving (i.e., $t^s = 1$) or rejecting (i.e.,

---

[1]Section B.2 provides further discussion of the shared primitive cost model and the log-normal assumption on agents' cost sensitivities.

$t^s = 0$) this customer's loan application (i.e., $x^s$), as some customers might be able to pay back the loan while others do not. In addition, we assume the random variable $Z$ represents exogenous noise that only affects the outcome $Y$. This captures external events that occur after the decision $t^s$ is made, such as unexpected income changes, unforeseen expenses, or temporary economic slowdown.

On the other hand, the agent does not modify their base features $x^b$ if either they receive a positive treatment at the first try, i.e., $t^b = 1$, or all recommended actions $x_j^r \in \mathcal{X}^r$ yield lower utility than retaining $x^b$, e.g., due to high modification costs (Vo et al., 2026). In these cases, we simply define $x^s := x^b$ and $t^s := t^b$. Consequently, their outcome is realized as $y := h(x^s, t^s, z) = h(x^b, t^b, z)$.

**DM's objective.** We define the value of a policy $\pi$ as $V(\pi) = \mathbb{E}_\pi[Y]$, which, for instance, reflects the expected profit of the DM from a loan application. Given a logging policy $\pi_0$ and the respective observational data[2] $D = \{(x_i^b, t_i^b, \mathcal{X}_i^r, x_i^s, t_i^s, y_i)\}_{i=1}^k$, the DM's goal is to estimate $V(\pi)$ of a target policy $\pi$ via an estimator $\hat{V}(\pi, D)$.

## 2.2. Policy Value Decomposition

The policy value $V(\pi)$ can be decomposed as follows:

$$
\begin{aligned}
V(\pi) = \mathbb{E}_\pi[Y] &= \sum_{t^s, x^s} \mathbb{E}_\pi[Y \mid t^s, x^s] \, p_\pi(t^s, x^s) \\
&= \sum_{t^s, x^s} \mathbb{E}_\pi[Y \mid t^s, x^s] \Big( \sum_{t^b, x^b} p_\pi(t^s \mid x^s, t^b, x^b) \\
&\qquad\qquad p_\pi(x^s \mid t^b, x^b) p_\pi(t^b, x^b) \Big) \\
&= \sum_{t^s, x^s, t^b, x^b} \underbrace{\mathbb{E}_\pi[Y \mid t^s, x^s]}_{①} \underbrace{p_\pi(t^s \mid x^s, t^b, x^b)}_{②} \\
&\qquad\qquad \underbrace{p_\pi(x^s \mid t^b, x^b)}_{③} \underbrace{p_\pi(t^b, x^b)}_{④}.
\end{aligned}
$$

In what follows, we explain each term in detail.

① **Conditional expected outcome** $\mathbb{E}_\pi[Y \mid t^s, x^s]$. This term models the expected outcome conditioned on the strategically updated covariate $x^s$ and treatment $t^s$. To further simplify it, we make the following assumption.

**Assumption 2.1** (Unconfoundedness). *The noise $Z$ and the treatment $T^s$ are conditionally independent given the strategically updated covariate $X^s$.*

This is a standard assumption in OPE literature (Uehara et al., 2020; Kallus et al., 2022) and trivially holds in our setting, by definition of $Z$. We note that in many real-world settings where the noise $Z$ is correlated with agents' features $X^s$, unconfoundedness does not hold trivially under strategic behavior. This is because any intervention on $T^s$

(by the DM) induces a corresponding intervention on $X^s$ by the strategic agent, over which the DM has no control; see, e.g., Munro (2025); Vo et al. (2024). Although strategic behavior introduces multiple challenges in OPE, our work focuses on the challenge of anticipating the covariate shift and assumes that the noise $Z$ is exogenous.

Under Assumption 2.1, we can drop the subscript $\pi$ from the conditional expected outcome as $\mathbb{E}[Y \mid t^s, x^s]$ becomes identifiable from observational data.

② **Strategic propensity** $p_\pi(t^s \mid x^s, t^b, x^b)$. This term captures an agent's chance to receive the treatment $t^s$, given a specific observation $\{x^s, t^b, b^b\}$. It largely depends on the DM's policy $\pi$ and can be computed straightforwardly.

③ **Strategic covariate shift** $p_\pi(x^s \mid t^b, x^b)$. This term captures the covariate shift arising from strategic behavior. We consider two cases of $t^b$. If $t^b = 1$, then by definition, the agent does not update their features, thus $p_\pi(x^s \mid T^b = 1, x^b) = \mathbb{1}(x^s = x^b)$. In contrast, when $t^b = 0$, we have

$$
③ = \sum_{e \in \mathcal{E}} p(x^s \mid T^b = 0, x^b, e) p_\pi(e \mid x^b). \tag{3}
$$

Given that the agent might have multiple utility maximizers, we present the following result to ensure that there is only one maximizer, almost surely. This allows us to avoid making assumptions about how the agent might break ties.

**Lemma 2.2** (Unique utility maximizer). *For any agent with $(x^b, t^b = 0)$ that receives recommendation set $\mathcal{X}^r$ (coming from the ARex $e$), if it holds that $d(x_\bullet^r, x^b) \neq d(x_\diamond^r, x^b)$ for any pair $x_\bullet^r \neq x_\diamond^r$ in $\mathcal{X}^r$, then*

$$
p_{\alpha \mid X^b}(|\arg\max_{x \in \mathcal{X}^f} u(\pi, \alpha, x, x^b)| = 1 \mid x^b) = 1. \tag{4}
$$

Lemma 2.2 says that when the DM provides recommendations to the agent, as long as there are no two feature updates with the same distance to $x^b$, then there is only a unique utility maximizer for this agent, almost surely. We use this result to further decompose $p(x^s \mid T^b = 0, x^b, e)$.

Let $\mathcal{X}_{-s}^f := \mathcal{X}^f \setminus \{x^s\}$ denote the set of feasible feature updates excluding the value $x^s$. Then, we have

$$
\begin{aligned}
&p(x^s \mid e, T^b = 0, x^b) \\
&= p(\{\pi(x^s) - \alpha d(x^s, x^b) > \pi(x) - \alpha d(x, x^b)\} \\
&\qquad \forall x \in \mathcal{X}_{-s}^f \mid e, T^b = 0, x^b) \, \mathbb{1}(x^s \in \mathcal{X}^f) \\
&= p(\{\pi(x^s) - \pi(x) > \alpha(d(x^s, x^b) - d(x, x^b))\} \\
&\qquad \forall x \in \mathcal{X}_{-s}^f \mid x^b) \, \mathbb{1}(x^s \in \mathcal{X}^f),
\end{aligned}
$$

where the strict inequality follows because $x^s$ is the unique utility maximiser a.s. and we can drop the conditioning variables $\{E, T^b\}$ because the rewritten expression contains only $\alpha$ as the source of randomness. In the following, we

write $\Delta_\pi(x, x') := \pi(x) - \pi(x')$ and $\Delta_d(x, x', x'') := d(x, x'') - d(x', x'')$ to simplify the notation.

Next, we define the three complementary sets for $\mathcal{X}^f_{-s}$. Let $\mathcal{X}^f_- := \{x : x \in \mathcal{X}^f_{-s} \ \& \ \Delta_d(x^s, x, x^b) < 0\}$ denote the subset of $\mathcal{X}^f_{-s}$ where the distances between its members to $x^b$ are larger than that of $x^s$ to $x^b$. We define $\mathcal{X}^f_+$ and $\mathcal{X}^f_0$ analogously where the former corresponds to the case $\Delta_d(x^s, x, x^b) > 0$ and the latter $\Delta_d(x^s, x, x^b) = 0$. We then define these three variables:

$$\delta^l := \max\left\{0, \ \max_{x \in \mathcal{X}^f_-}\left\{\frac{\Delta_\pi(x^s, x)}{\Delta_d(x^s, x, x^b)}\right\}\right\},$$

$$\delta^u := \min_{x \in \mathcal{X}^f_+}\left\{\frac{\Delta_\pi(x^s, x)}{\Delta_d(x^s, x, x^b)}\right\}, \quad \delta^0 := \min_{x \in \mathcal{X}^f_0} \Delta_\pi(x^s, x).$$

If $\mathcal{X}^f_-$ and/or $\mathcal{X}^f_+$ are empty, we can set $\delta^l := 0$ and $\delta^u := \infty$. If $\mathcal{X}^f_0$ is empty, we can equivalently set $\delta^0$ to some arbitrarily positive value. We can then rewrite $p(x^s \mid e, T^b = 0, x^b)$ as

$$
\begin{aligned}
&p(x^s \mid e, T^b = 0, x^b) \\
&= p(\{\Delta_\pi(x^s, x) > \alpha\Delta_d(x^s, x, x^b)\} \ \forall x \in \mathcal{X}^f_{-s} \mid x^b) \\
&\qquad \mathbb{1}(x^s \in \mathcal{X}^f) \\
&= p\Big(\frac{\Delta_\pi(x^s, x)}{\Delta_d(x^s, x, x^b)} > \alpha \ \forall x \in \mathcal{X}^f_+ \\
&\qquad \& \ \frac{\Delta_\pi(x^s, x)}{\Delta_d(x^s, x, x^b)} < \alpha \ \forall x \in \mathcal{X}^f_- \\
&\qquad \& \ \Delta_\pi(x^s, x) > 0 \ \forall x \in \mathcal{X}^f_0 \mid x^b\Big)\mathbb{1}(x^s \in \mathcal{X}^f) \\
&= p\Big(\max_{x \in \mathcal{X}^f_-}\frac{\Delta_\pi(x^s, x)}{\Delta_d(x^s, x, x^b)} < \alpha < \min_{x \in \mathcal{X}^f_+}\frac{\Delta_\pi(x^s, x)}{\Delta_d(x^s, x, x^b)} \\
&\qquad \& \ \min_{x \in \mathcal{X}^f_0} \Delta_\pi(x^s, x) > 0 \mid x^b\Big)\mathbb{1}(x^s \in \mathcal{X}^f) \\
&= p(\{\delta^l < \alpha < \delta^u\} \ \& \ \{\delta^0 > 0\} \mid x^b)\mathbb{1}(x^s \in \mathcal{X}^f) \\
&= p(\{\delta^l < \alpha < \delta^u\} \mid x^b)\mathbb{1}(\delta^0 > 0)\mathbb{1}(x^s \in \mathcal{X}^f),
\end{aligned}
$$

where the second to last equation follows from our definitions of $\delta^l, \delta^u, \delta^0$ and because $\alpha$ follows a log-normal distribution. Therefore, being able to evaluate the CDF of $\alpha$ allows the DM to estimate $p_\pi(x^s \mid t^b, x^b)$ and anticipate the strategic covariate shift.

④ **Non-strategic joint distribution** $p_\pi(t^b, x^b)$. This term is not influenced by the agents' strategic behavior and can be computed straightforwardly. We discuss the estimation for $p(x^b)$ in the next section.

The next subsection provides an estimation procedure for the unknown parameters $\theta = (\beta, \beta_0, \sigma)$ of the cost model.

## 2.3. Learning the Agents' Cost Model

Given each observation of agents' strategic behavior, i.e., a tuple $(x^s, \mathcal{X}^r, t^b = 0, x^b)$, we define the per-observation likelihood contribution as

$$f(\delta^l, \delta^u, x^b; \theta) := p_\theta\Big(\delta^l_{(x^s, e, x^b)} < \alpha < \delta^u_{(x^s, e, x^b)} \mid x^b\Big),$$

if $\delta^l_{(x^s, e, x^b)} < \delta^u_{(x^s, e, x^b)}$. When $\delta^l_{(x^s, e, x^b)} = \delta^u_{(x^s, e, x^b)}$, we set $f(\delta^l, \delta^u, x^b; \theta) := c$ for some constant $c \in (0, 1)$.

Given $n$ observations $\{(x^s_i, e_i, t^b_i = 0, x^b_i)\}^n_{i=1}$ collected from agents who received negative base treatment, i.e, $t^b_i = 0$, under the logging policy $\pi_0$, we define the empirical log-likelihood function as

$$\mathcal{Q}_n(\theta) = \frac{1}{n}\sum_{i=1}^n \ln f(\delta^l_i, \delta^u_i, x^b_i; \theta),$$

and denote $\hat{\theta}_n := \arg\max_{\theta \in \Theta} \mathcal{Q}_n(\theta)$ as the maximum likelihood estimator of the agents' cost model. We then present the conditions for this estimator to be consistent.

**Assumption 2.3** (Compact parameter space). The parameter space $\Theta \subset \mathbb{R}^p \times \mathbb{R} \times \mathbb{R}^+$ is compact.

**Assumption 2.4** (Finite covariates). The space of covariate vectors $\mathcal{X} \subset \mathbb{R}^d$ is finite.

**Assumption 2.5** (Weak positivity & full-rank design matrix). There exists a subset $\mathcal{X}^b_\diamond \subseteq \mathcal{X}$ of size at least $p+1$ such that

- for each $x^b \in \mathcal{X}^b_\diamond$, there exist two positive values $\delta^{u1} \neq \delta^{u2}$ where the two observations $(0, \delta^{u1}, x^b)$ and $(0, \delta^{u2}, x^b)$ occur with positive probabilities, i.e.,

$$p(\delta^l = 0, \delta^u = \delta^{u1}, X^b = x^b) > 0,$$
$$p(\delta^l = 0, \delta^u = \delta^{u2}, X^b = x^b) > 0;$$

- the augmented design matrix $\tilde{\Phi}_{x^b}$ has full column rank, where we define

$$\tilde{\Phi}_{x^b} = \begin{bmatrix} 1 & \phi(x^b_1)^\top \\ \vdots & \vdots \\ 1 & \phi(x^b_{p+1})^\top \end{bmatrix} \quad \forall x^b_1, \dots, x^b_{p+1} \in \mathcal{X}^b_\diamond.$$

The first part of Assumption 2.5 imposes a weak positivity condition, requiring sufficient variation in the observed strategic responses. The second part is likewise mild, as it depends primarily on the marginal $P_{X^b}$ and the mapping $\phi$.

**Theorem 2.6** (Consistency of $\hat{\theta}_n$). *Let $\theta^* = (\beta^*, \beta^*_0, \sigma^*)$ be the true parameters of agents' cost model. Under Assumptions 2.3–2.5 and that the DM recommends covariate updates with different distances to $x^b$ (Lemma 2.2), $\hat{\theta}_n \xrightarrow{p} \theta^*$.*

Based on these results, standard off-policy evaluation estimators can be derived. In the following section, we present a doubly robust estimator.

# 3. Strategy-Robust Doubly Robust Estimator

In this section, we introduce a doubly robust estimator for off-policy evaluation that explicitly accounts for strategic covariate shift. We refer to this estimator as the **strategy-robust doubly robust (SDR)** estimator:

$$\hat{V}_{\text{SDR}}(\pi) = \hat{V}_{\text{S-IPS-res}}(\pi) + \hat{V}_{\text{S-DM}}(\pi),$$

where $\hat{V}_{\text{S-IPS-res}}$ refers to the (strategy-robust) inverse propensity score (IPS)-based estimate of the residual and $\hat{V}_{\text{S-DM}}$ the (strategy-robust) direct method estimator of the policy value. In particular, they correspond to

$$\hat{V}_{\text{S-IPS-res}}(\pi) = \frac{1}{m} \sum_{i=1}^{m} \left( y_i - \hat{\mu}(t_i^s, x_i^s) \right) \frac{\hat{p}_\pi(t_i^s, x_i^s, t_i^b \mid x_i^b)}{\hat{p}_{\pi_0}(t_i^s, x_i^s, t_i^b \mid x_i^b)},$$

$$\hat{V}_{\text{S-DM}}(\pi) = \sum_{\mathcal{T} \times \mathcal{X} \times \mathcal{T} \times \mathcal{X}} \hat{\mu}(t^s, x^s) \hat{p}_\pi(t^s, x^s, t^b, x^b),$$

where $\hat{\mu}(t^s, x^s)$ denote the estimator of the conditional expected outcome $\mu(t^s, x^s) := \mathbb{E}[Y \mid t^s, x^s]$. The importance weights can also be defined as:

$$w(t^s, x^s, t^b, x^b) := \frac{p_\pi(t^s, x^s, t^b \mid x^b)}{p_{\pi_0}(t^s, x^s, t^b \mid x^b)},$$

whose empirical version $\hat{w}(t^s, x^s, t^b, x^b)$ can be obtained by replacing $p_\pi$ and $p_{\pi_0}$ with their empirical estimates $\hat{p}_\pi$ and $\hat{p}_{\pi_0}$, respectively.

**Assumption 3.1** (Overlap)**.** Given the logging policy $\pi_0$ and the evaluation policy $\pi$, we assume that $p_\pi(t^s, x^s, t^b | x^b) > 0$ implies $p_{\pi_0}(t^s, x^s, t^b | x^b) > 0$, for all values $(t^s, x^s, t^b, x^b) \in \mathcal{T} \times \mathcal{X} \times \mathcal{T} \times \mathcal{X}$.

The density $\hat{p}(x^b)$ can be estimated via empirical frequency, since the space $\mathcal{X}$ is finite. Under the i.i.d sampling assumption, this estimator is consistent by the law of large numbers. We next establish the double robustness property of $\hat{V}_{\text{SDR}}$: the estimator is consistent if either the outcome model $\hat{\mu}(x^s, t^s)$ is consistent or Assumption 3.1 holds.

**Theorem 3.2** (Consistency of $\hat{V}_{\text{SDR}}$)**.** *Suppose that the estimator $\hat{p}_\pi(t^s, x^s, t^b, x^b)$ is consistent for any $\pi$, the nuisance components (i.e., $\hat{w}, \hat{\mu}$) are estimated independently, and the samples for $\hat{V}_{\text{S-IPS-res}}$ are collected separately, then $\hat{V}_{\text{SDR}}$ is consistent if either the outcome model $\hat{\mu}(x^s, t^s)$ is consistent or Assumption 3.1 (overlap) holds.*

# 4. Experiments

We conduct experiments on a synthetic dataset and a real-world dataset to verify our theoretical results. Firstly, we show that as the sample size increases, our estimates for (i) the parameters of the agents' cost model and for (ii) the policy value converge in probability towards the ground-truth, reflecting the consistency results in Theorems 2.6

and 3.2. Secondly, we show that a standard doubly robust approach, without correct adjustment for the strategic covariate shift, will produce incorrect policy value estimates. This demonstrates the importance of correctly adjusting for the strategic covariate shift. Thirdly, we investigate a scenario where there are some agents who behave differently from our assumed behavioral model in Section 2.1. This is to understand how such deviation from our behavioral assumption can bias the policy value estimates.

## 4.1. Synthetic Data

We generate $N \geq 1000$ agents with 2-dimensional observable feature vectors $x^b \in \mathcal{X} = \{-10, \ldots, 10\}^2 \subset \mathbb{Z}^2$. Each agent has a cost function of the form $c(x, x^b) := 0.05 \times \alpha \|x - x^b\|_2^2$ where $\ln \alpha \sim \mathcal{N}(\beta^\top \phi(x^b) + \beta_0, \sigma^2)$. The true parameters are $\beta^* = [1.0, 1.2]^\top$, $\beta_0^* = 0.5$, and $\sigma^* = 1.0$. In addition, any strategic movement outside of the space $\mathcal{X}$ results in an infinite cost. The outcome function is $h(x^s, t^s) := ([5, 5]^\top x^s)t^s + 5$. The DM aims to estimate the conditional expected outcome $\mathbb{E}[Y \mid t^s, x^s]$ via a predictive model $\hat{\mu} : \mathcal{T} \times \mathcal{X} \to \mathcal{Y}$.

For a logistic function $g(a) := 1/(1 + \exp(-a))$, we consider the two logging policies $\pi_0^{\text{strict}}$ and $\pi_0^{\text{lax}}$ such that $\pi_0^{\text{strict}}(x) = g([4, 4]^\top x)$ and $\pi_0^{\text{lax}}(x) = g([1, 1]^\top x)$ where the $\pi_0^{\text{strict}}$ has stronger discriminative power (i.e., closer to being a deterministic policy). We use it to study the case where Assumption 3.1 (overlap) is violated. These two logging policies will later induce two different logged datasets $D_{\pi_0^{\text{strict}}}$ and $D_{\pi_0^{\text{lax}}}$. We use the former to evaluate our SDR estimator when there is limited overlap and the latter to evaluate when $\hat{\mu}$ is misspecified.

The DM uses a deterministic explanation policy $\tau$ such that for any agent with base covariates $x^b$, they receive $\mathcal{X}^r = \{x^b + [0, 1]^\top, x^b + [1, 3]^\top, x^b + [1, 4]^\top\}$. The DM's goal is to estimate the value of the new treatment policy $\pi$ where $\pi(x) = g([1, 1]^\top x - 1)$.

**Baseline & evaluation.** We use our SDR estimator, with *mis-specified* cost model's parameters $\theta$, as the baseline (i.e., wrong adjustment for the covariate shift). In particular, we set the parameters as $\beta = [1.5, 0.8]^\top$, $\beta_0 = 0.2$, and $\sigma = 0.7$. We repeat the experiment multiple times, while increasing the number of agents $N$ from 1000 to 11000 with a step size of 500. For each $N$, we randomly generate a logged dataset of size $N$, then estimate the parameters of agents' cost model using the maximum likelihood approach and compute the SDR estimator. We repeat 30 times for each choice of $N$ to collect 30 noisy estimates.

**Result 1: All agents behave rationally.** Figure 2 shows the convergence of the estimates for the log-normal distribution of $\alpha$. The shrinkage of shaded regions implies the concentration of the noisy estimates around the ground-truth

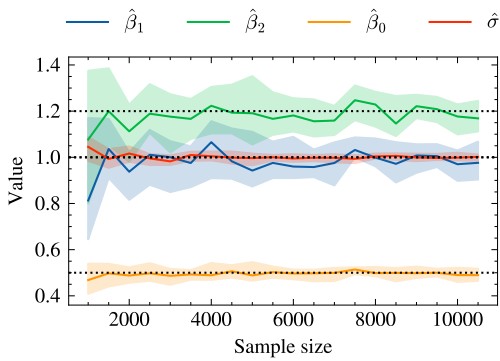

*Figure 2.* The plots show the consistency of the estimated parameters of the cost model (on the synthetic dataset). Each line shows the median of the noisy estimates. Each shaded region captures the estimates between $25\% - 75\%$ quantiles.

values, as the dataset size increases. This demonstrates the consistency of the estimated parameter vector $\hat{\theta}$. Figure 3 shows the convergence of our SDR estimator in several cases. Similarly, the shrinkage of the shaded regions shows the concentration of the noisy estimates around the ground-truth. This demonstrates the consistency of our SDR estimator, while the baseline produces incorrect estimates due to wrong assumption about agents' behavior.

**Result 2: Some agents behave irrationally.** We also test the robustness of our approach when agents behave differently from our assumed model in Section 2.1. In particular, we fix the number of "irrational" agents to 1000 and observe how the bias introduced by such irrationality goes down as the dataset size increases. For any agent who fails to modify their base covariate vector $x^b$ to any recommended value $x^r$, this agent may choose an arbitrary value $x$ drawn uniformly from the covariate space $\mathcal{X}$. Figure 4 shows the convergence of our SDR estimator when there are 1000 irrational agents. As the dataset size increases, our estimator converges as the number of samples from rational agents dominate. Since, in our LID setting, the DM can distinguish between agents' rational and irrational responses (i.e., by checking if $x^s \in \mathcal{X}^f$), our MLE procedure (for estimating the agents' cost model) simply discards these data points from irrational agents. This irrational behaviour is correlated with the cost sensitivity of agents, hence creating selection bias when the data is discarded.

### 4.2. German Credit Data

We preprocess the German credit dataset (Hofmann, 1994) following Xie & Zhang (2024). In particular, we exclude two sensitive attributes and retain 18 features, among which 8 are considered modifiable by strategic agents. The original dataset contains 1000 samples. We use CTGAN (Xu et al., 2019) to generate 200000 additional samples for our experiments. We use covariates from the German Credit dataset and simulate agents' strategic behavior and outcomes ac-

cording to our structural model.

**The outcome model.** Although the German Credit dataset provides a binary credit-risk label, our target outcome is the bank's realized profit, which is naturally continuous-valued. Accordingly, we define the synthetic outcome function as $Y := 10((\vec{1})^\top X^s)T + 5$, where the first term represents treatment-dependent loan profit and the constant term represents a processing fee. We evaluate the SDR estimator under both correctly specified and misspecified outcome model $\hat{\mu}$. In particular, a correctly specified model has the multiplicative form $\hat{\mu}(t^s, x^s) = \zeta_1^\top x^s + \zeta_2^\top t^s + \zeta_3((\vec{1})^\top x^s)t + \zeta_0$, while the misspecified model omits the interaction term.

**DM-agents interactions.** To set up the logging policy $\pi_0$ and evaluation policy $\pi$, we first fit a logistic regression model on the original German Credit binary labels and use the fitted model as a base scoring function. We then scale its coefficients to obtain policies with different levels of selectiveness. Suppose that the base policy is $\pi_{\text{base};\eta}(x) = 1/(1+\exp(-(\eta_0+\eta_1^\top x))$ where $\eta = (\eta_0, \eta_1)$. Then, the logging and evaluation policies are respectively $\pi_0 := \pi_{\text{base};0.1\eta}$ and $\pi := \pi_{\text{base};2\eta}$.

To operationalize ARexes, we let the DM generate all possible candidates for feature updates $x^r$ whose Hamming distance (to the base covariate vector) is equal to one, i.e., $d_{\text{Hamming}}(x^r, x^b) = 1$. Then, for each agent, the DM gives top 3 recommendations $x^r$ with the highest weighted scores $\pi(x^r)/d(x^r, x^b)$. Such recommendations aim to balance the benefit and the (base) effort necessary for an agent to adapt. For example, a bank might recommend a change in a customer's profile that benefits the customer without incurring high cost for them.

**The agents' behavioral model.** Similar to Vo et al. (2026), we set up an agent's cost function as

$$c(x, x') = \alpha d(x, x') = \alpha 0.001 \sum_{i \in \mathcal{I}} \frac{|x_i - x_i^b|}{|x_i^U - x_i^L|},$$

where $\mathcal{I}$ is the set of indices of the 8 modifiable features and $[x_i^U, x_i^L]$ denotes the valid range of a feature $x_i$. Any change in non-modifiable features incurs infinite cost.

Recall that $\alpha \sim \mathcal{N}(\beta^\top \phi(x^b) + \beta_0, \ \sigma^2)$. We apply PCA on the semi-synthetic dataset containing the agents' base covariate vectors $x^b$, extract the top 4 principle components, and use them to construct a PCA transformation $\phi : \mathcal{X} \to \mathbb{R}^4$. We set the true parameters to $\beta^* = [0.25, 0.25, 0.25, 0.25]^\top$, $\beta_0^* = 0.5$, and $\sigma^* = 1.0$.

**Estimation & evaluation.** To account for continuous features, we use Monte Carlo approximation to compute the SDR (policy-value) estimate. In particular, we do not fit a model to predict $p(x^b)$, but compute only $\hat{p}(t^s, x^s, t^b|x^b)$ and use it to weight the data points when computing the SDR estimate. We use our SDR estimator, with *mis-*

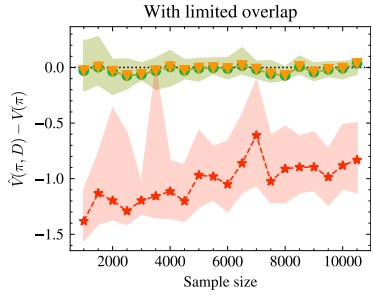

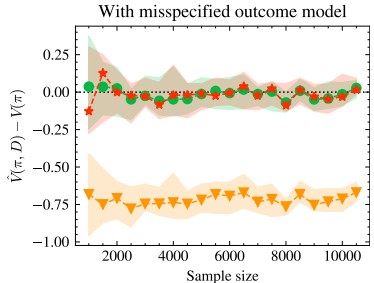

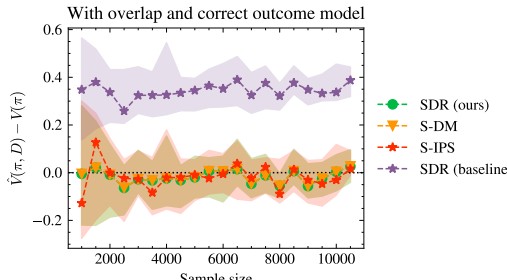

*(a)* Despite limited overlap, our SDR estimator remains consistent, whereas the IPS-based estimator fails.

*(b)* When $\hat{\mu}$ is misspecified, our SDR estimator remains consistent, whereas the DM estimator fails.

*(c)* The baseline SDR approach produces incorrect estimates when it assumes wrong agents' strategic behavior.

*Figure 3.* Illustrations of the differences between the estimates for policy value and the ground-truth (on the synthetic dataset). Each line shows the median of the errors of the noisy estimates. Each shaded region captures the errors within the $25\% - 75\%$ quantiles.

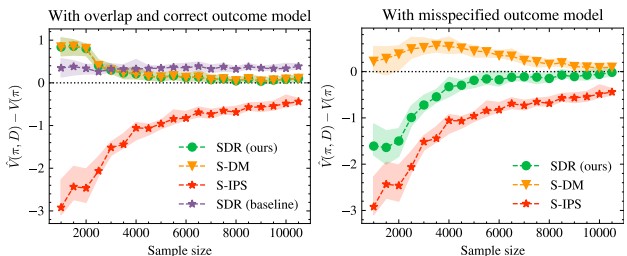

*Figure 4.* Similar to Figure 3, these plots show the errors of the noisy policy-value estimates, but when there are 1000 agents who behave irrationally (on the synthetic dataset). This demonstrates that although such irrational behaviour biases our estimates, as we obtain more samples from rational agents, the bias tends to zero.

*specified* cost model's parameters $\theta$, as the baseline. In particular, we set the cost model's parameters as $\beta = [0.225, 0.225, 0.225, 0.225]^\top$, $\beta_0 = 0.2$, and $\sigma = 0.7$. We repeat the experiment multiple times, while increasing the number of agents $N$ from 9500 to 190000 with a step size of 9500. Similar to the synthetic data case, for each $N$, we repeat the experiment 30 times to obtain 30 noisy estimates.

Figure 5 shows the convergence of our estimate $\hat{\theta}$ of the log-normal distribution's parameters and the convergence of our SDR estimator. The shrinkage of shaded regions implies the concentration of the noisy estimates around the ground-truth values, as the dataset size increases.

## 5. Related Work

**(Off-)policy evaluation and policy learning.** Existing work on OPE under covariate shift focuses on *exogenous* shifts, where changes in the covariate distribution do not arise in response to the policy being evaluated. For example, Ue-hara et al. (2020); Guo et al. (2024) study settings in which the test distribution of covariates is assumed to be known, while Kallus et al. (2022) assumes the test distribution of covariates falls within a pre-specified set. These assumptions allow the evaluation problem to condition on a fixed or externally specified target distribution. In contrast, when covariate shifts are policy-dependent, the test distribution varies with the policy being evaluated and is therefore neither known nor constrained to a policy-invariant set. As a result, existing approaches, which rely on exogenous specification of the target distribution, are not suitable in this setting. Another line of work studies optimizing decision policies and typically deals with strategic behavior through repeated interactions (Perdomo et al., 2020; Munro, 2025; Chen et al., 2024; Perdomo, 2025). While leveraging repeated online interactions can help adapt policies or infer agents' responses over time, this is often infeasible in high-stakes settings where experiments are costly or ethically constrained. This makes them unsuitable for OPE.

**Strategic machine learning.** A prominent line of work that examines strategic behavior comes from strategic classification literature and its variants. While many of them rely on repeated online interactions (Shavit et al., 2020; Harris et al., 2022b; Horowitz & Rosenfeld, 2023; Vo et al., 2024; Xie & Zhang, 2024), several considers offline settings (Hardt et al., 2016; Levanon & Rosenfeld, 2021; Rosenfeld & Rosenfeld, 2024). However, these works in offline settings typically assume that agents' responses to policy changes can be modeled precisely, commonly formalized through a *single* and *known* cost function. This strong assumption limits the applicability of their approaches to OPE. Closest to our work in the spirit of relaxing such strong assumption is the work of Rosenfeld & Rosenfeld (2024), which considers a one-shot setting without knowing the exact agents' cost function. However, there are two main distinctions. Their approach relies on an externally specified set of feasible cost functions, which cannot capture the heterogeneity of agents' behavior. Secondly, their focus is on optimizing the classifier for the worst-case scenario while our focus is on estimating the performance of a policy.

**Contract design.** The information loss induced by global disclosure is closely related to classical problems of infor-

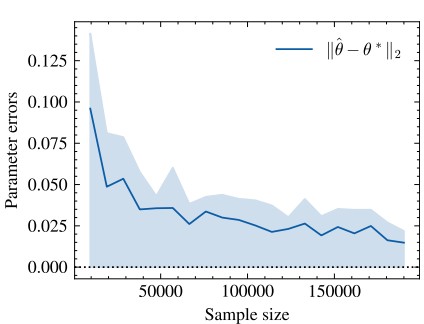
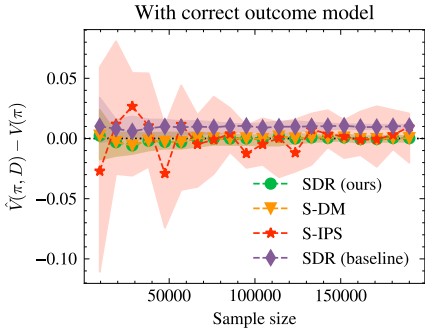
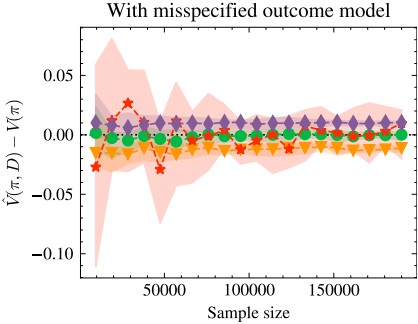

*(a)* The errors of our estimates $\hat{\theta}$ decay as the sample size increases.

*(b)* Baseline SDR gives incorrect estimates under the wrong behavioral assumption.

*(c)* Our SDR estimator is consistent when $\hat{\mu}$ is misspecified, unlike the DM estimator.

*Figure 5.* These plots illustrate the convergence of our estimators on the German credit dataset. Each line shows the median of the errors of the noisy estimates. Each shaded region captures the errors within the $25\% - 75\%$ quantiles.

mation asymmetry studied in economics, such as adverse selection in contract design (Rothschild & Stiglitz, 1976; Bolton & Dewatripont, 2004). While our setting differs substantially from standard economic models in both objectives and setup, a common insight is that the principal's choice of how interactions are structured plays a central role in mitigating information asymmetry. In particular, screening models (Baron & Myerson, 1982) emphasize that a principal can deliberately design an interaction scheme so that agents' responses reveal information that would otherwise remain unobserved. Our perspective draws on this high-level idea: local information disclosure represents a design choice that structures interactions so that agents' pre-strategic covariates are observed to the DM, thereby preserving information that is lost under global disclosure.

**Recourse and explanation design.** Our use of the action recommendation-based explanation (ARex) framework (Vo et al., 2026) as a local disclosure mechanism is related to the algorithmic recourse and counterfactual explanations literature, which studies how decision makers can provide individuals with actionable feedback to achieve a desired decision outcome (see Section 2 and Karimi et al. (2021); Wachter et al. (2018)). Several works model how releasing such feedback can and induce strategic responses, creating feedback loops between the decision rule and the observed covariates (Tsirtsis & Gomez Rodriguez, 2020) and potentially rendering explanations *performative* (König et al., 2026). Recent work also stresses that recourse should be *set-valued*: offering multiple counterfactuals can better heterogeneous user preferences (Mothilal et al., 2020).

## 6. Conclusion and Discussion

When agents behave strategically, it gives rise to a policy-dependent covariate shift, affecting the existing OPE approaches that rely on a fixed or externally specified covariate distribution. While agents' responses can be anticipated

precisely with full knowledge of their behavioral model, such an assumption rarely holds in practice. Our work is not only among the first to examine the problem of OPE under strategic behavior, but also proposes an approach that does not assume full knowledge of the agents' behavioral model, unlike much of related work in strategic machine learning.

In summary, we extend the ARex framework (Vo et al., 2026) specifically to the task of OPE under strategic behavior. We then propose an estimation procedure to learn the parameters of agents' cost model. Finally, we construct a strategy-robust doubly robust estimator and prove its consistency. Beyond these technical contributions, our work draws attention to the issue of information asymmetry when GID is assumed in the presence of strategic behaviour. When agents modify their features strategically in response to the DM's policy, two key challenges emerge.

The first challenge is the *policy-dependent* covariate shift, which the DM must account for to accurately estimate policy performance. In many applications, LID is a design choice that can be leveraged to elicit additional information from strategic agents. We demonstrate this using ARexes. More broadly, this suggests a new approach for mitigating information asymmetry in learning problems, enabling the DM to relax strong assumptions about agent behavior.

The second challenge is the breakdown of unconfoundedness, a standard assumption in OPE. While treatment assignment $T$ may be unconfounded with the outcome $Y$ given covariates $X$ in non-strategic settings, this assumption can fail under strategic behavior (see Section 2.2). Consequently, $\mathbb{E}[Y \mid t, x]$ may no longer be identifiable. Although not the main focus, our framework can be extended to infer latent variables governing strategic adaptations $X^s$, analogous to the approach in Section 2.3. Conditioning on such latent information may help block confounding paths between $X^s$ and $Y$. Crucially, this becomes possible through the use of LID to reveal additional information about agents.

## Acknowledgements

We sincerely thank the members of the Rational Intelligence (RI) Lab, including Anurag Singh, Joseph Sheils, and Monseej Purkayastha for their insightful discussions, constructive feedback, and invaluable contributions to this work. We also thank the anonymous reviewers for their valuable feedback to improve our work. Kiet Q. H. Vo is a doctoral candidate at Saarland University.

## Impact Statement

This paper presents work whose goal is to advance the field of Machine Learning. There are many potential societal consequences of our work, none which we feel must be specifically highlighted here.

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

# A. Summary of Notations

Table 1 presents a summary of the notation we used in this paper.

*Table 1.* Summary of the notations

| Notation | Meaning |
|---:|---|
| $X^b$ | A Random variable denoting an agent's base/original/pre-strategic covariates |
| $x^b$ | A value taken by the random variable $X^b$ |
| $\mathcal{X}$ | Space of covariates |
| $\pi : \mathcal{X} \to [0, 1]$ | A DM's decision policy |
| $T^b$ | A random variable denoting the treatment assigned to an agent with base covariates $X^b$ |
| $t^b$ | A value taken by the random variable $T^b$ |
| $\mathcal{T} = \{0, 1\}$ | Binary treatment space |
| $\mathcal{X}^r = \{x_j^r\}_{j=1}^k$ | The set of $k$ recommendations |
| $e = \{(x_j^r), \pi(x_j^r)\}_{j=1}^k$ | Explanation containing recommendations and policy outcomes |
| $\mathcal{X}^f := \{x_b\} \cup \mathcal{X}^r$ | A set of feasible actions containing base covariates and recommendations |
| $c : \mathcal{X} \times \mathcal{X} \to \mathbb{R}_{\geq 0}$ | Cost function denoting the cost involved in modifying covariates |
| $d : \mathcal{X} \times \mathcal{X} \to \mathbb{R}_{\geq 0}$ | A (primitive cost) function known to DM and common to all agents |
| $\alpha \in (0, \infty)$ | Agent's specific cost sensitivity |
| $Z, Y$ | Random variables denoting the exogenous noise and the agent's outcome |
| $z, y$ | Values taken by the random variables $Z, Y$ |
| $\mathcal{Z}, \mathcal{Y}$ | Spaces of values for $z, y$ |

# B. Additional Discussion

## B.1. On the Agents' Behavioral Model

The agents' behavioral model in Equation (1) of Section 2.1 follows prior work on strategic learning under local information disclosure (Tsirtsis & Gomez Rodriguez, 2020; Vo et al., 2026). In particular, the motivation comes from the ARex framework (Vo et al., 2026), where the authors argue that, in many real-world settings, the DM can collect additional information (e.g., through surveys) to generate recommendations better aligned with agents' preferences, thereby reducing the likelihood that agents deviate from the recommended actions. Our work further extends the ARex framework to allow an unrestricted number of recommendations, which can further mitigate the chance that agents deviate from the presented recommendation set.

Recent work in strategic classification has questioned whether agents necessarily behave according to idealized best-response models. However, many such works operate under substantially different explanation settings. For example, in Ebrahimi et al. (2025), agents are provided feature importance weights and must infer for themselves how feature modifications translate into utility gains. As discussed by Vo et al. (2026), such uncertainty, which is common in many explanation paradigms, can give rise to misinterpretation. In contrast, ARexes directly present actionable recommendations together with their associated policy outcomes, thereby substantially reducing ambiguity about the utility consequences of recommended actions.

Nevertheless, as with most strategic learning frameworks, our approach still relies on a stylized behavioral model of agents. Although ARexes can be made more practical through additional preference elicitation and our extension mitigates the likelihood that agents deviate from the recommendation set, such behavior cannot be ruled out entirely. Extending OPE under strategic behavior to accommodate richer or boundedly rational response models remains an important direction for future work. In Section 4.1, we additionally provide a simple experiment to study how deviations from the assumed behavioral model may affect estimation performance.

## B.2. On the Agents' Cost Model

**The cost structure.** The cost model in Section 2.1 separates the modification cost $c(x, x')$ into two components: a shared primitive cost $d(x, x')$ and an agent-specific sensitivity parameter $\alpha$. Intuitively, $d(x, x')$ captures the burden imposed

by society (e.g., time, effort, and monetary expenses) required to modify one's covariates. In contrast, $\alpha$ reflects how a specific agent internalizes this burden in their utility, translating the primitive cost $d(x, x')$ into a personal cost $c(x, x')$ that is weighed against the chance of receiving a positive treatment.

The current problem formulation can in principle be generalized to accommodate richer forms of heterogeneous cost functions. For example, instead of a scalar sensitivity parameter $\alpha$, one could consider feature-specific sensitivities that allow different agents to find some feature modifications easier or harder than others. This could correspond to replacing the scalar scaling factor with a quadratic form such as $(x - x')^\top \Lambda (x - x')$, where the matrix $\Lambda$ captures how different feature modifications are weighted. However, such extensions substantially complicate the analysis of the induced strategic covariate shift $p(x^s | t^b, x^b)$, since one would need to derive the distribution of scalar costs $(x - x')^\top \Lambda (x - x')$ from distributions over random vector- or matrix-valued latent variables $\Lambda$. Developing estimation procedures for such settings remains a promising direction for future work.

**Modeling the cost sensitivity.**   The modeling choice for the cost sensitivity $\alpha$ reflects that agents with different characteristics, captured by $x^b$, may differ in their willingness to bear effort or monetary expenses. The transformation $\phi(x^b)$ allows the conditional mean of $\ln \alpha$ to depend on covariates while accommodating mixtures of categorical and numerical variables. In contrast, the variance parameter $\sigma^2$ captures residual heterogeneity arising from latent factors outside $x^b$, such as temporary personal constraints or unobserved motivational differences. Using a shared variance parameter therefore reflects the assumption that, while different groups may differ in their average attitudes toward effort, the dispersion around those averages is broadly comparable across groups.

The log-normal model has a natural motivation based on the central limit theorem (CLT): $\alpha > 0$ is a positive scale parameter that describes an agent's overall cost sensitivity, which can be interpreted as the aggregate effect of many latent factors (e.g., liquidity, time constraints, opportunity cost, motivation, risk tolerance, etc.). If these factors combine approximately multiplicatively on the original scale, then $\ln(\alpha) \mid x^b$ becomes approximately additive. If the latent contributions additionally have finite variance, standard central-limit-type arguments motivate approximating $\ln(\alpha) \mid x^b$ by a normal distribution.

We model the conditional mean of $\ln \alpha$ as a linear function of transformed covariates for interpretability and statistical tractability. Since the DM observes only one interaction per agent, the available data are substantially more limited than in repeated-interaction settings, motivating the use of a parsimonious parametric model while still allowing nonlinear relationships through the transformation $\phi(x^b)$.

## C. Uniqueness of the Agent's Utility Maximizer

Here, we prove Lemma 2.2 under a parametric assumption on $\alpha$ and a design condition for $\mathcal{X}^r$.

Because we assume the cost function to have the form $C(x, x') := \alpha d(x, x')$ where $\alpha \in \mathbb{R}^+$, when the DM designs $\mathcal{X}^r$ (for each agent) such that all recommended actions $x_j^r \in \mathcal{X}^r$ have different primitive costs $d(x_j^r, x^b)$, there is only at most one value for $\alpha$ that could result in multiple utility maximizers.

To see this, we first suppose that there are two utility maximizers $x_\bullet^r, x_\diamond^r \in \mathcal{X}^r$ for an agent $i$ such that $d(x_\bullet^r, x^b) \neq d(x_\diamond^r, x^b)$. Note that this is possible because we assume the DM knows $d$. Then, for this agent, the following must hold:

$$u(\pi, \alpha_i, x_\bullet^r, x^b) = u(\pi, \alpha_i, x_\diamond^r, x^b) \tag{5}$$

$$\Rightarrow \pi(x_\bullet^r) - \alpha_i d(x_\bullet^r, x^b) = \pi(x_\diamond^r) - \alpha_i d(x_\diamond^r, x^b) \tag{6}$$

$$\Rightarrow \pi(x_\bullet^r) - \pi(x_\diamond^r) = \alpha_i \big( \underbrace{d(x_\bullet^r, x^b) - d(x_\diamond^r, x^b)}_{\neq 0} \big), \tag{7}$$

which only holds for at most one value of $\alpha_i$.

Given $(\pi, \alpha, \mathcal{X}^f, x^b)$, the event $\{| \arg\max_{x \in \mathcal{X}^f} u(\pi, \alpha, x, x^b)| \geq 2\}$ is equivalent to a set of finite $\alpha$ values $A = \{\alpha' : | \arg\max_{x \in \mathcal{X}^f} u(\pi, \alpha', x, x^b)| \geq 2\}$.

When we assume $\ln \alpha \mid x^b \sim \mathcal{N}(\cdot, \cdot)$, we have $p(\alpha \in A | x^b) = 0$ for any set $A$ containing finite values of $\alpha$. Therefore, Lemma 2.2 holds as

$$p_{\alpha | X^b} \Big( | \arg\max_{x \in \mathcal{X}^f} u(\pi, \alpha, x, x^b)| \geq 2 \mid x^b \Big) = 0. \tag{8}$$

Note that $g$ and $\mathcal{X}^f$ are arguments to evaluate above quantity. This concludes the proof.

### C.1. Probability of the Agent's Response

Here, we show how to arrive at strict inequalities of utility comparison, by using the result from Lemma 2.2.

Let $M_{\mathcal{X}}$ denote the set of utility maximizers for an agent, we have the following, from previous result:

$$p\big(|M_{\mathcal{X}}| = 1 \mid \mathcal{X}^r, T^b = 0, x^b\big) = 1 \quad \& \quad p\big(|M_{\mathcal{X}}| \geq 2 \mid \mathcal{X}^r, T^b = 0, x^b\big) = 0. \tag{9}$$

Note that in our setup, $\alpha \perp\!\!\!\perp \{\mathcal{X}^r, T^b\} \mid X^b$. Let $W := \mathbb{1}_{|M_{\mathcal{X}}|=1}$ be a binary random variable denoting if the set of maximizers has the size of 1 or not. We have

$$p(W = 1 \mid \mathcal{X}^r, T^b = 0, x^b) = 1 \quad \& \quad p(W = 0 \mid \mathcal{X}^r, T^b = 0, x^b) = 0. \tag{10}$$

Then,

$$p(x^s | \mathcal{X}^r, T^b = 0, x^b) \tag{11}$$

$$= \sum_{w \in \{0,1\}} p\big(\{\text{agent picks } x^s\} \ \& \ W = w \mid \mathcal{X}^r, T^b = 0, x^b\big) \, \mathbb{1}\big(x^s \in \mathcal{X}^f\big) \tag{12}$$

$$= \sum_{w \in \{0,1\}} p\big(\{\text{agent picks } x^s\} \mid W = w, \dots\big) \underbrace{p\big(W = w \mid \mathcal{X}^r, T^b = 0, x^b\big)}_{=0 \text{ if } w=0} \, \mathbb{1}\big(x^s \in \mathcal{X}^f\big) \tag{13}$$

$$= p\big(\{\text{agent picks } x^s\} \ \& \ W = 1 \mid \mathcal{X}^r, T^b = 0, x^b\big) \, \mathbb{1}\big(x^s \in \mathcal{X}^f\big) \tag{14}$$

$$= p\big(\{\text{agent picks } x^s\} \ \& \ |M_{\mathcal{X}}| = 1 \mid \mathcal{X}^r, T^b = 0, x^b\big) \, \mathbb{1}\big(x^s \in \mathcal{X}^f\big) \tag{15}$$

$$= p\big(\{u(\pi, C, x^s, x^b) > u(g, C, x, x^b)\} \, \forall x \in \mathcal{X}^f \setminus \{x^s\} \mid \mathcal{X}^r, T^b = 0, x^b\big) \, \mathbb{1}\big(x^s \in \mathcal{X}^f\big) \tag{16}$$

$$= p\big(\{u(\pi, C, x^s, x^b) > u(g, C, x, x^b)\} \, \forall x \in \mathcal{X}^f \setminus \{x^s\} \mid x^b\big) \, \mathbb{1}\big(x^s \in \mathcal{X}^f\big). \tag{17}$$

## D. Estimation of Agents' Strategic Responses

Here, we prove the consistency of the maximum likelihood estimator $\hat{\theta}_n$. Recall that

$$\ln \alpha \mid x^b \sim \mathcal{N}(\beta^\top \phi(x^b) + \beta_0, \sigma^2), \tag{18}$$

where $\phi : \mathcal{X} \to \mathbb{R}^p$ denotes some known transformation function of the covarite vector $x^b$. Furthermore, the conditional CDF, $F(\alpha|x^b; \theta)$, is parameterized by $\theta = (\beta, \beta_0, \sigma) \in \Theta \subseteq \mathbb{R}^p \times \mathbb{R} \times \mathbb{R}^+$.

Given each observation of agents' strategic behavior, i.e., a tuple $(x^s, \mathcal{X}^r, t^b = 0, x^b)$, we define the per-observation likelihood contribution as

$$f(\delta^l, \delta^u, x^b; \theta) = \begin{cases} p_\theta\Big(\delta^l_{(x^s, \mathcal{X}^r, x^b)} < \alpha < \delta^u_{(x^s, \mathcal{X}^r, x^b)} \mid x^b\Big) & \text{if } 0 \leq \delta^l_{(x^s, \mathcal{X}^r, x^b)} < \delta^u_{(x^s, \mathcal{X}^r, x^b)}, \\ f_{\text{const}} & \text{otherwise}, \end{cases} \tag{19}$$

where any choice for the constant $f_{\text{const}} \in (0, 1)$ works and each pair of $(\delta^l, \delta^u)$ is the transformation of an observation $(x^s, \mathcal{X}^r, t^b = 0, x^b)$, as defined in the main paper. Note that $0 \leq \delta^l \leq \delta^u$ by construction, so $f$ outputs $f_{\text{const}}$ when $\delta^l = \delta^u$.

Note that in our log-normal model, in the special case $0 = \delta^l < \delta^u$, we have

$$f(0, \delta^u, x^b; \theta) = p_\theta(0 < \alpha < \delta^u \mid x^b) = p_\theta(\alpha < \delta^u \mid x^b) \tag{20}$$

$$= F_{\alpha|X^b}(\delta^u | x^b; \theta). \tag{21}$$

Given $n$ observations $\{(x_i^s, \mathcal{X}_i^r, t_i^b = 0, x_i^b)\}_{i=1}^n$ collected from agents who received negative base treatment, i.e, $t_i^b = 0$, we define the empirical log-likelihood and its population version as follows:

$$\mathcal{Q}_n(\theta) = \frac{1}{n} \sum_{i=1}^n \ln f(\delta_i^l, \delta_i^u, x_i^b; \theta), \tag{22}$$

$$\mathcal{Q}(\theta) = \mathbb{E}_{P_{\delta^l, \delta^u, X^b|T^b}} \left[ \ln f(\delta^l, \delta^u, X^b; \theta) \,\Big|\, T^b = 0 \right], \tag{23}$$

where the distribution $P_{\delta^l, \delta^u, X^b|T^b}$ is induced by $P_{X^s, \mathcal{X}^r, X^b|T^b}$. Furthermore, the density of an observation can be expressed as

$$p(x^s, \mathcal{X}^r, x^b \mid T^b = 0) = p(x^s \mid \mathcal{X}^r, x^b, T^b = 0)\, p(\mathcal{X}^r, x^b \mid T^b = 0) \tag{24}$$

$$= p(\delta^l < \alpha < \delta^u \mid x^b \,;\, \theta_0)\, \mathbb{1}(x^s \in \mathcal{X}^r \cup \{x^b\})\, p(\mathcal{X}^r, x^b \mid T^b = 0), \tag{25}$$

where the last line follows from what we derive in the main paper, and of course, from Lemma 2.2. Similarly, assuming the log-normal model for $\alpha$ and the uniqueness of utility maximiser (almost surely), the same decomposition holds for any arbitrary parameter value $\theta$, i.e.

$$p(x^s, \mathcal{X}^r, x^b \mid T^b = 0 \,;\, \theta) = p(x^s \mid \mathcal{X}^r, x^b, T^b = 0 \,;\, \theta)\, p(\mathcal{X}^r, x^b \mid T^b = 0) \tag{26}$$

$$= p(\delta^l < \alpha < \delta^u \mid x^b \,;\, \theta)\, \mathbb{1}(x^s \in \mathcal{X}^r \cup \{x^b\})\, p(\mathcal{X}^r, x^b \mid T^b = 0). \tag{27}$$

We will use this decomposition in the proof for Lemma D.5 (unique likelihood maximiser) later.

Let $\hat{\theta}_n := \arg\max_{\theta \in \Theta} \mathcal{Q}_n(\theta)$ and $\theta_0$ is the true parameter value in $F(\alpha|x^b; \theta)$, our goal is to prove that $\hat{\theta}_n \xrightarrow{p} \theta_0$.

We introduce the following lemma that will help proving subsequent results.

**Lemma D.1** (Continuity). *For each tuple* $(\delta^l, \delta^u, x^b) \in \mathbb{R}^{d+2}$, *the corresponding functions* $f(\delta^l, \delta^u, x^b; \theta)$ *and* $\ln f(\delta^l, \delta^u, x^b; \theta)$ *are continuous w.r.t.* $\theta \in \mathbb{R}^p \times \mathbb{R} \times \mathbb{R}^+$.

*Proof.* We expand the function $f$ for the case of $0 < \delta^l < \delta^u$:

$$p_\theta \left( \delta^l_{(x^s, \mathcal{X}^r, x^b)} < \alpha < \delta^u_{(x^s, \mathcal{X}^r, x^b)} \mid x^b \right) \tag{28}$$

$$= F_{\alpha|X^b}(\delta^u|x^b; \theta) - F_{\alpha|X^b}(\delta^l|x^b; \theta) \tag{29}$$

$$= \frac{1}{2} \left[ \operatorname{erf}\left( \frac{\ln \delta^u - \beta^\top \phi(x^b) - \beta_0}{\sigma \sqrt{2}} \right) - \operatorname{erf}\left( \frac{\ln \delta^l - \beta^\top \phi(x^b) - \beta_0}{\sigma \sqrt{2}} \right) \right], \tag{30}$$

where $\operatorname{erf}$ denotes the error function.

Because each member function inside $f$ is continuous w.r.t. $\theta$, such as inversion $(1/\sigma)$, linear transformation $(\beta^\top \phi(x^b) + \beta_0)$, and $\operatorname{erf}(\cdot)$, their composition is also continuous on $\Theta$. Similar argument applies for the case of $0 = \delta^l < \delta^u$.

For the case of collapsed intervals, $f$ outputs a constant so it is continuous. Thus $f$ is continuous on $\Theta$.

Similarly, as $\ln$ is continuous on the domain $\mathbb{R}^+$, $\ln \circ f$ is continuous on $\Theta$. This concludes the proof. $\qquad \square$

## D.1. Consistency of $\hat{\theta}_n$

**Lemma D.2** (Dominance). *Under Assumption 2.3, Assumption 2.4, and when Lemma 2.2 holds, there exists an upper bound* $\eta \in \mathbb{R}$ *such that*

$$|\ln f(\delta^l, \delta^u, x^b, \theta)| < \eta \quad \forall \theta \in \Theta, \,\forall (\delta^l, \delta^u, x^b) \in \mathcal{C}_{g,\sigma,d,\mathcal{X}}, \tag{31}$$

*where* $\mathcal{C}_{g,\sigma,d,\mathcal{X}}$ *denotes the set of all possible values of* $(\delta^l, \delta^u, x^b)$ *induced by* $\{g, \sigma, d, \mathcal{X}\}$.

*Proof.* From Lemma D.1, for each configuration $(\delta^l, \delta^u, x^b)$, the function $f(\delta^l, \delta^u, x^b; \theta)$ is continuous w.r.t. $\theta$. In addition, because the space $\Theta$ is compact, $f(\delta^l, \delta^u, x^b; \theta)$ is bounded on $\Theta$, by the extreme value theorem.

Because the space $\mathcal{X}$ is finite, $\mathcal{C}_{g,\sigma,d,\mathcal{X}}$ is also finite, for a given instantiation of $\{g, \sigma, d\}$. Consequently, there are finitely many bounds for $f(\delta^l, \delta^u, x^b; \theta)$ for all $(\delta^l, \delta^u, x^b) \in \mathcal{C}_{g,\sigma,d,\mathcal{X}}$ and $\theta \in \Theta$. Therefore, there exists a constant $\eta \in \mathbb{R}$ such that

$$|\ln f(\delta^l, \delta^u, x^b, \theta)| < \eta \quad \forall \theta \in \Theta, \,\forall (\delta^l, \delta^u, x^b) \in \mathcal{C}_{g,\sigma,d,\mathcal{X}}. \tag{32}$$

This concludes the proof. $\qquad \square$

**Lemma D.3** (Uniform convergence). *Under Assumption 2.3 and Assumption 2.4, we have*

$$\sup_{\theta \in \Theta} |\mathcal{Q}_n(\theta) - \mathcal{Q}(\theta)| \xrightarrow{p} 0. \tag{33}$$

*Proof.* From Assumption 2.3 (compactness), Lemma D.1 (continuity), and Lemma D.2 (dominance), we directly get the desired result, by using Lemma 2.4 of Newey & McFadden (1994). $\square$

**Lemma D.4** (Identifiability). *Under Assumption 2.3, Assumption 2.4, Assumption 2.5, and when Lemma 2.2 holds, we have*

$$\left\{ f(\delta^l, \delta^u, x^b; \theta) = f(\delta^l, \delta^u, x^b; \theta_0) \quad \forall (\delta^l, \delta^u, x^b) \in \mathcal{C}_{g,\sigma,d,\mathcal{X}} \right\} \Rightarrow \theta = \theta_0. \tag{34}$$

*Proof.* Suppose that there exists a parameter value $\theta_\bullet$ such that $f(\delta^l, \delta^u, x^b; \theta_\bullet) = f(\delta^l, \delta^u, x^b; \theta_0)$ for all $(\delta^l, \delta^u, x^b) \in \mathcal{C}_{g,\sigma,d,\mathcal{X}}$. From Assumption 2.5 (weak positivity & and full-rank), there exist two observations $(0, \delta^{u1}, x^b)$ and $(0, \delta^{u2}, x^b)$, for some $x^b \in \mathcal{X}$, such that for all $\delta^u \in \{\delta^{u1}, \delta^{u2}\}$,

$$f(0, \delta^u, x^b; \theta_\bullet) = f(0, \delta^u, x^b; \theta_0) \tag{35}$$

$$\Rightarrow \mathrm{erf}\left( \frac{\ln \delta^u - \beta(\theta_\bullet)^\top \phi(x^b) - \beta_0(\theta_\bullet)}{\sigma(\theta_\bullet)\sqrt{2}} \right) = \mathrm{erf}\left( \frac{\ln \delta^u - \beta(\theta_0)^\top \phi(x^b) - \beta_0(\theta_0)}{\sigma(\theta_0)\sqrt{2}} \right), \tag{36}$$

where we use $\beta(\theta), \beta_0(\theta), \sigma(\theta)$ to denote the respective elements in a parameter vector $\theta$. Because the error function is strictly increasing on the domain $\mathbb{R}$, hence invertible, we get the following for all $\delta^u \in \{\delta^{u1}, \delta^{u2}\}$:

$$\frac{\ln \delta^u - \beta(\theta_\bullet)^\top \phi(x^b) - \beta_0(\theta_\bullet)}{\sigma(\theta_\bullet)} = \frac{\ln \delta^u - \beta(\theta_0)^\top \phi(x^b) - \beta_0(\theta_0)}{\sigma(\theta_0)} \tag{37}$$

$$\Rightarrow \ln \delta^u \left( \frac{1}{\sigma(\theta_\bullet)} - \frac{1}{\sigma(\theta_0)} \right) + \frac{-\beta(\theta_\bullet)^\top \phi(x^b) - \beta_0(\theta_\bullet)}{\sigma(\theta_\bullet)} - \frac{-\beta(\theta_0)^\top \phi(x^b) - \beta_0(\theta_0)}{\sigma(\theta_0)} = 0. \tag{38}$$

Because the above holds for two different values of $\delta^u \in \mathbb{R}^+$, it must hold that $\sigma(\theta_\bullet) = \sigma(\theta_0)$, then we obtain

$$\beta(\theta_\bullet)^\top \phi(x^b) + \beta_0(\theta_\bullet) = \beta(\theta_0)^\top \phi(x^b) + \beta_0(\theta_0) \tag{39}$$

$$\Rightarrow \left( \beta(\theta_\bullet) - \beta(\theta_0) \right)^\top \phi(x^b) + \left( \beta_0(\theta_\bullet) - \beta_0(\theta_0) \right) = 0. \tag{40}$$

When this scenario holds for $p + 1$ distinct values of $x^b$ such that the augmented design matrix $\tilde{\Phi}_{x^b}$ has full column rank (Assumption 2.5), by the closed form solution for ordinary least squares, we get $\beta(\theta_\bullet) - \beta(\theta_0) = 0$ and $\beta_0(\theta_\bullet) - \beta_0(\theta_0) = 0$. Thus $\theta_\bullet = \theta_0$ and this concludes the proof. $\square$

**Lemma D.5** (Unique likelihood maximizer). *Under Assumption 2.3, Assumption 2.4, Assumption 2.5, and additionally when Lemma 2.2 holds, the true parameter $\theta_0$ is the unique global maximizer of $\mathcal{Q}(\theta)$.*

*Proof.* We use the idea in Lemma 2.2 of Newey & McFadden (1994) where the strict version of Jensen's inequality is employed for a non-constant random variable. For any $\theta \neq \theta_0$,

$$\mathcal{Q}(\theta_0) - \mathcal{Q}(\theta) = \mathbb{E}_{P_{\delta^l, \delta^u, X^b | T^b}} \left[ \ln \frac{f(\delta^l, \delta^u, X^b; \theta_0)}{f(\delta^l, \delta^u, X^b; \theta)} \,\Big|\, T^b = 0 \right] \tag{41}$$

$$= \mathbb{E}_{P_{\delta^l, \delta^u, X^b | T^b}} \left[ -\ln \frac{f(\delta^l, \delta^u, X^b; \theta)}{f(\delta^l, \delta^u, X^b; \theta_0)} \,\Big|\, T^b = 0 \right] \tag{42}$$

$$> -\ln \mathbb{E}_{P_{\delta^l, \delta^u, X^b | T^b}} \left[ \frac{f(\delta^l, \delta^u, X^b; \theta)}{f(\delta^l, \delta^u, X^b; \theta_0)} \,\Big|\, T^b = 0 \right], \tag{43}$$

where $f(\delta^l, \delta^u, X^b; \theta) / f(\delta^l, \delta^u, X^b; \theta_0)$ is a non-constant random variable with support in $\mathcal{C}_{g,\sigma,d,\mathcal{X}}$ thanks to Lemma D.4 (identifiability) and Lemma 2.2 (unique utility maximiser), which results in non-zero probabilities for non-degenerate intervals.

We further rewrite the inequality below, where variables $\delta^l, \delta^u$ are just some transformations of $x^s, \mathcal{X}^r, x^b$ which in turn are transformations of $\alpha, \mathcal{X}^r, x^b$,

$$\mathcal{Q}(\theta_0) - \mathcal{Q}(\theta) > -\ln \mathbb{E}_{P_{\delta^l, \delta^u, X^b | T^b}} \left[ \frac{f(\delta^l, \delta^u, X^b; \theta)}{f(\delta^l, \delta^u, X^b; \theta_0)} \,\Big|\, T^b = 0 \right] \tag{44}$$

$$= -\ln \mathbb{E}_{P_{X^s, \mathcal{X}^r, X^b | T^b}} \left[ \frac{f(\delta^l, \delta^u, X^b; \theta)}{f(\delta^l, \delta^u, X^b; \theta_0)} \,\Big|\, T^b = 0 \right] \tag{45}$$

$$= -\ln \int_{\mathcal{X} \times \mathcal{P}(\mathcal{X}) \times \mathcal{X}} \frac{f(\delta^l, \delta^u, x^b; \theta)}{f(\delta^l, \delta^u, x^b; \theta_0)} \, p(x^s, \mathcal{X}^r, x^b \mid T^b = 0; \theta_0) \, d(x^s, \mathcal{X}^r, x^b) \tag{46}$$

$$= -\ln \int_{\mathcal{X} \times \mathcal{P}(\mathcal{X}) \times \mathcal{X}} \frac{p(\delta^l < \alpha < \delta^u \mid x^b; \theta)}{p(\delta^l < \alpha < \delta^u \mid x^b; \theta_0)} \, p(x^s, \mathcal{X}^r, x^b \mid T^b = 0; \theta_0) \, d(x^s, \mathcal{X}^r, x^b) \tag{47}$$

$$= -\ln \int_{\mathcal{X} \times \mathcal{P}(\mathcal{X}) \times \mathcal{X}} p(x^s, \mathcal{X}^r, x^b \mid T^b = 0; \theta) \, d(x^s, \mathcal{X}^r, x^b) \tag{48}$$

$$= -\ln 1 \tag{49}$$

$$= 0. \tag{50}$$

Note that we could cancel out the term $f(\delta^l, \delta^u, x^b; \theta_0)$ by using our earlier decomposition for $p(x^s, \mathcal{X}^r, x^b \mid T^b = 0; \theta_0)$ at the beginning of this section. Furthermore, we do not have to worry about the case where $f(\delta^l, \delta^u, x^b; \theta_0) = f_{\text{const}}$ because the probability of observing a degenerate interval is zero, thanks to Lemma 2.2.

Thus, we have shown that $\theta_0$ is the unique global maximiser for $\mathcal{Q}$. This concludes the proof. $\qquad\square$

We now show the main theorem on the consistency of $\hat{\theta}_n$.

**Theorem D.6** (Consistency of $\hat{\theta}_n$). *Under Assumption 2.3, Assumption 2.4, Assumption 2.5, and when Lemma 2.2 holds, we have $\hat{\theta}_n \xrightarrow{p} \theta_0$.*

*Proof.* From Assumption 2.3 (compactness), Lemma D.1 (continuity), Lemma D.3 (uniform convergence), and Lemma D.5 (unique likelihood maximizer), we directly get the desired result, by using Theorem 2.1 of Newey & McFadden (1994). $\quad\square$

### D.2. Consistency of $F(\alpha|x^b; \hat{\theta}_n)$

Because we assume the log-normal model where $\ln \alpha \mid x^b \sim \mathcal{N}(\beta^\top \phi(x^b) + \beta_0, \sigma^2)$, we get

$$F(\alpha|x^b; \hat{\theta}_n) = \frac{1}{2} \left[ 1 + \text{erf} \left( \frac{\ln \alpha - \hat{\beta}^\top \phi(x^b) - \hat{\beta}_0}{\hat{\sigma}\sqrt{2}} \right) \right], \tag{51}$$

where $\hat{\theta} = (\hat{\beta}, \hat{\beta}_0, \hat{\sigma})$.

From the proof of Lemma D.1, the mapping $F(\alpha|x^b; \theta)$ (for any given pair of values $\{\alpha, x^b\}$) is continuous on $\Theta$ and with $p(\theta_0 \in \Theta) = 1$, then by the continuous mapping theorem (Van der Vaart, 2000) we get

$$\left( \hat{\theta}_n \xrightarrow{p} \theta_0 \right) \Rightarrow \left( F(\alpha|x^b; \hat{\theta}_n) \xrightarrow{p} F(\alpha|x^b; \theta_0) \right) \quad \forall (\alpha, x^b) \in \mathbb{R}^+ \times \mathcal{X}. \tag{52}$$

Similarly, $p(x^s \mid \mathcal{X}^r, x^b, T^b = 0; \theta)$ is continuous on $\Theta$ because the other multiplicative terms are constant w.r.t. $\theta$. We then obtain consistency of $p(x^s \mid \mathcal{X}^r, x^b, T^b = 0; \hat{\theta}_n)$.

## E. Strategy-Robust Doubly Robust Estimator

We use $\pi$ to denote the evaluation policy, $\pi_0$ the logging policy, and $\hat{\mu}(x^s, t^s)$ the regression-based estimator for the conditional expected outcome $\mathbb{E}[Y|x^s, t^s]$.

$$\hat{V}_{\text{SDR}}(\pi) = \hat{V}_{\text{S-IPS-res}}(\pi) + \hat{V}_{\text{S-DM}}(\pi) \tag{53}$$

$$= \frac{1}{m} \sum_{i=1}^{m} \left(y_i - \hat{\mu}(x_i^s, t_i^s)\right) \frac{p_\pi(t_i^s|x_i^s, t_i^b, x_i^b)}{p_{\pi_0}(t_i^s|x_i^s, t_i^b, x_i^b)} \frac{\hat{p}_\pi(x_i^s|t_i^b, x_i^b)}{\hat{p}_{\pi_0}(x_i^s|t_i^b, x_i^b)} \frac{p_\pi(t_i^b|x_i^b)}{p_{\pi_0}(t_i^b|x_i^b)} \tag{54}$$

$$+ \sum_{\mathcal{X}^2 \times \mathcal{T}^2} \hat{\mu}(x^s, t^s) p_\pi(t^s|x^s, t^b, x^b) \hat{p}_\pi(x^s|t^b, x^b) p_\pi(t^b|x^b) \hat{p}(x^b). \tag{55}$$

Note that we assume the above important weights are well-defined on the observed samples, which is mild because both the samples and the denominator terms are obtained under the same logging policy $\pi_0$.

For ease of presentation, we define the following terms:

$$w(t^s, x^s, t^b, x^b) := \frac{p_\pi(t^s|x^s, t^b, x^b)}{p_{\pi_0}(t^s|x^s, t^b, x^b)} \frac{p_\pi(x^s|t^b, x^b)}{p_{\pi_0}(x^s|t^b, x^b)} \frac{p_\pi(t^b|x^b)}{p_{\pi_0}(t^b|x^b)}, \tag{56}$$

$$\hat{w}(t^s, x^s, t^b, x^b) := \frac{p_\pi(t^s|x^s, t^b, x^b)}{p_{\pi_0}(t^s|x^s, t^b, x^b)} \frac{\hat{p}_\pi(x^s|t^b, x^b)}{\hat{p}_{\pi_0}(x^s|t^b, x^b)} \frac{p_\pi(t^b|x^b)}{p_{\pi_0}(t^b|x^b)}. \tag{57}$$

We then use $w_i$ and $\hat{w}_i$ to refer to $w(t_i^s, x_i^s, t_i^b, x_i^b)$ and $\hat{w}(t_i^s, x_i^s, t_i^b, x_i^b)$, respectively.

In addition, we define the estimated densities:

$$\hat{p}_\pi(t^s, x^s, t^b, x^b) := p_\pi(t^s|x^s, t^b, x^b) \, \hat{p}_\pi(x^s|t^b, x^b) \, p_\pi(t^b|x^b) \, \hat{p}(x^b), \tag{58}$$

$$\hat{p}_\pi(t^s, x^s) := \sum_{\mathcal{X} \times \mathcal{T}} \hat{p}_\pi(t^s, x^s, t^b, x^b), \tag{59}$$

and let $\hat{P}_{T^s, X^s; \pi}$ denote the distribution that corresponds to the density $\hat{p}_\pi(t^s, x^s)$.

We rewrite the SDR estimator as

$$\hat{V}_{\text{SDR}}(\pi) = \hat{V}_{\text{S-IPS-res}}(\pi) + \hat{V}_{\text{S-DM}}(\pi) \tag{60}$$

$$= \frac{1}{m} \sum_{i=1}^{m} \left(y_i - \hat{\mu}(x_i^s, t_i^s)\right) \hat{w}_i + \sum_{\mathcal{X}^2 \times \mathcal{T}^2} \hat{\mu}(x^s, t^s) \hat{p}_\pi(t^s, x^s, t^b, x^b) \tag{61}$$

$$= \frac{1}{m} \sum_{i=1}^{m} \left(y_i - \hat{\mu}(x_i^s, t_i^s)\right) \hat{w}_i + \sum_{\mathcal{X} \times \mathcal{T}} \hat{\mu}(x^s, t^s) \hat{p}_\pi(t^s, x^s) \tag{62}$$

$$= \frac{1}{m} \sum_{i=1}^{m} \left(y_i - \hat{\mu}(x_i^s, t_i^s)\right) \hat{w}_i + \mathbb{E}_{\hat{P}_{T^s, X^s; \pi}} \left[\hat{\mu}(X^s, T^s)\right]. \tag{63}$$

# F. Consistency of $\hat{V}_{\text{SDR}}$

We show the double robustness property for $\hat{V}_{\text{SDR}}$. When $\hat{p}_\pi(t^s, x^s, t^b, x^b)$ is consistent (for any $\pi$, including the case $\pi = \pi_0$), the nuisance components (i.e., $\hat{w}$, $\hat{\mu}$) are estimated independently, and the samples for $\hat{V}_{\text{S-IPS-res}}$ are collected separately, $\hat{V}_{\text{SDR}}$ is consistent if either $\hat{\mu}(x^s, t^s)$ is consistent or Assumption 3.1 (overlap) holds (i.e., $w(\cdot)$ is well defined).

## F.1. When $\hat{\mu}(x^s, t^s)$ Is Consistent

We inject the term $\mu(x_i^s, t_i^s)$ into $\hat{V}_{\text{SDR}}(\pi)$ as follows:

$$\hat{V}_{\text{SDR}}(\pi) = \frac{1}{m} \sum_{i=1}^{m} \left(y_i - \hat{\mu}(x_i^s, t_i^s)\right) \hat{w}_i + \mathbb{E}_{\hat{P}_{T^s, X^s; \pi}} \left[\hat{\mu}(X^s, T^s)\right] \tag{64}$$

$$= \underbrace{\frac{1}{m} \sum_{i=1}^{m} \left(y_i - \mu(x_i^s, t_i^s)\right) \hat{w}_i}_{A} + \underbrace{\frac{1}{m} \sum_{i=1}^{m} \left(\mu(x_i^s, t_i^s) - \hat{\mu}(x_i^s, t_i^s)\right) \hat{w}_i}_{B} + \underbrace{\mathbb{E}_{\hat{P}_{T^s, X^s; \pi}} \left[\hat{\mu}(X^s, T^s)\right]}_{C}. \tag{65}$$

When $\hat{\mu}(x^s, t^s)$ and $\hat{p}_\pi(t^s, x^s, t^b, x^b)$ are consistent estimators, then by the continuous mapping theorem (Van der Vaart, 2000), $C \xrightarrow{p} \mathbb{E}_{P_{T^s, X^s; \pi}}[\mu(X^s, T^s)]$, which means $C \xrightarrow{p} V(\pi)$.

When $\hat{w}$ is a fixed function (which can be achieved by estimated from a separate data set), we can use the law of large numbers to show that $A \xrightarrow{p} 0$. In particular, we have

$$\mathbb{E}_{g_0}\left[\left(Y - \mu(X^s, T^s)\right)\hat{w}(T^s, X^s, T^b, X^b)\right] \tag{66}$$

$$= \mathbb{E}_{g_0}\left[\mathbb{E}\left[\left(Y - \mu(X^s, T^s)\right)\hat{w}(T^s, X^s, T^b, X^b) \mid T^s, X^s, T^b, X^b\right]\right] \tag{67}$$

$$= \mathbb{E}_{g_0}\left[\underbrace{\mathbb{E}\left[Y - \mu(X^s, T^s) \mid T^s, X^s, T^b, X^b\right]}_{=0 \text{ from the definition of } \mu}\hat{w}(T^s, X^s, T^b, X^b)\right] \tag{68}$$

$$= 0, \tag{69}$$

where $\hat{w}(T^s, X^s, T^b, X^b)$ can be moved outside of the conditional expectation term because $\hat{w}(T^s, X^s, T^b, X^b)$ is a constant when conditioned on $\{T^s, X^s, T^b, X^b\}$, which comes from the fact that $\hat{w}$ is a non-random function.

As the above expectation exists, we can use the law of large numbers and show that $A \xrightarrow{p} \mathbb{E}_{\pi_0}\left[\left(Y - \mu(X^s, T^s)\right)\hat{w}(T^s, X^s, T^b, X^b)\right]$, which gives $A \xrightarrow{p} 0$.

Similarly, in the B term, when $\hat{\mu}(x^s, t^s)$ is a consistent estimator, it means $\hat{\mu}(x^s, t^s) - \mu(x^s, t^s) \xrightarrow{p} 0$. Consequently, this gives us $B \xrightarrow{p} 0$ as long as $\hat{w}(\cdot)$ is bounded. This boundedness behaviour of $\hat{w}$ can be obtained when $\hat{w}$ and $\hat{\mu}$ are estimated independently from separate datasets and the space $\mathcal{X} \times \mathcal{T}$ is finite.

Together, we have $\hat{V}_{\text{SDR}} \xrightarrow{p} V(\pi)$.

### F.2. When Overlap Holds

We inject the terms $w_i$ into $\hat{V}_{\text{SDR}}$ as follows:

$$\hat{V}_{\text{SDR}}(g) = \frac{1}{m}\sum_{i=1}^{m}\left(y_i - \hat{\mu}(x_i^s, t_i^s)\right)\hat{w}_i + \mathbb{E}_{\hat{P}_{T^s, X^s; \pi}}[\hat{\mu}(X^s, T^s)] \tag{70}$$

$$= \frac{1}{m}\sum_{i=1}^{m}y_i w_i + \frac{1}{m}\sum_{i=1}^{m}y_i\left(\hat{w}_i - w_i\right) + \frac{1}{m}\sum_{i=1}^{m}\hat{\mu}(x_i^s, t_i^s)\left(w_i - \hat{w}_i\right) \tag{71}$$

$$- \frac{1}{m}\sum_{i=1}^{m}\hat{\mu}(x_i^s, t_i^s)w_i + \mathbb{E}_{\hat{P}_{T^s, X^s; \pi}}[\hat{\mu}(X^s, T^s)]. \tag{72}$$

We further denote the following for ease of presentation:

$$A := \frac{1}{m}\sum_{i=1}^{m}y_i w_i, \tag{73}$$

$$B := \frac{1}{m}\sum_{i=1}^{m}y_i\left(\hat{w}_i - w_i\right), \tag{74}$$

$$C := \frac{1}{m}\sum_{i=1}^{m}\hat{\mu}(x_i^s, t_i^s)\left(w_i - \hat{w}_i\right), \tag{75}$$

$$D := -\frac{1}{m}\sum_{i=1}^{m}\hat{\mu}(x_i^s, t_i^s)w_i + \mathbb{E}_{\hat{P}_{T^s, X^s; \pi}}[\hat{\mu}(X^s, T^s)], \tag{76}$$

where $\hat{V}_{\text{SDR}} = A + B + C + D$ and we will show that $A \xrightarrow{p} V(\pi)$ while the remaining terms converge to 0.

Using the law of large numbers, we have $A \xrightarrow{p} \mathbb{E}_{\pi_0}[Yw(T^s, X^s, T^b, X^b)]$, where $\mathbb{E}_{\pi_0}[Yw(T^s, X^s, T^b, X^b)] = \mathbb{E}_\pi[Y] = V(\pi)$ if the overlap assumption holds.

When $\hat{w}_i$ is a consistent estimator of $w_i$, by using the continuous mapping theorem (Van der Vaart, 2000), we get $B \xrightarrow{p} 0$ and $C \xrightarrow{p} 0$, as long as the mapping $\hat{\mu}$ is fixed (e.g., by estimating from a separate dataset).

When the mapping $\hat{\mu}$ is fixed, by the law of large numbers, the first term in $D$ converges to $\mathbb{E}_{P_{T^s, X^s; \pi}}[-\hat{\mu}(X^s, T^s)]$. When $\hat{p}_\pi(t^s, x^s, t^b, x^b)$ is consistent, the second term in $D$ converges to $\mathbb{E}_{P_{T^s, X^s; \pi}}[\hat{\mu}(X^s, T^s)]$. Thus, together $D \xrightarrow{p} 0$.

Putting everything together, $\hat{V}_{\text{SDR}}(\pi) \xrightarrow{p} V(\pi)$.

