# OpenReview forum: "Off-Policy Evaluation with Strategic Agents via Local Disclosure"
_ICML.cc/2026/Conference — ICML 2026 regular_

### Official Review · Reviewer_V8eb · 2026-03-09

**Soundness:** 3
**Presentation:** 3
**Significance:** 3
**Originality:** 3
**Overall Recommendation:** 3
**Confidence:** 3

**Summary:**

This paper studies the problem of offline policy evaluation when there are strategic individuals. The author points out that traditional OPE usually assumes that the distribution of covariates does not change with the policy to be evaluated, but when individuals actively adjust their observable characteristics in order to obtain more favorable decisions, the covariate distribution drift induced by the policy will occur, thus making the standard OPE method invalid. To solve this problem, the paper introduces the LID mechanism, which enables decision-makers to observe the original characteristics of individuals before their strategic adjustment, and model individual response behavior based on this additional information. Specifically, the author models the individual's cost sensitivity as a conditional lognormal distribution, gives the consistency estimation method of the corresponding parameters, and further constructs a strategy robust double robust estimator that explicitly modifies the strategic covariate shift to estimate the value of the target policy. In theory, the paper proves consistency of the cost model parameter estimator and of the final policy value estimator; In experiments, the author verifies that the proposed method can obtain more accurate estimates than the standard DR method that ignores policy behavior in synthetic data. On the whole, the paper focuses on a relatively novel problem with realistic motivation, and tries to give a solution from the perspective of combining interaction mechanism design with OPE.

**Compliance With Llm Reviewing Policy:**

Affirmed.

**Key Questions For Authors:**

1. The core of the paper relies on LID mechanism to observe pre strategic covariates. Can the author explain more specifically in what real applications this setting is natural and achievable, and if the pre strategic covariates cannot be observed, is there any applicable version of this method that can be degenerated?

2. The method relies more on specific behavior model and conditional lognormal distribution hypothesis. Can the author supplement the analysis of model mismatch, for example, when individuals are not strictly optimized in the recommendation set, or the cost sensitivity distribution deviates from log normal, how will the estimated performance change?

3. The current experiment is all synthetic data. Does the author consider adding stronger stress test or experiments closer to the real scene to support the practical applicability of the method?

**Limitations:**

The main limitation of this paper is that its applicability largely depends on whether the LID interaction settings are realistic and feasible. At the same time, the method has strong structural assumptions on the individual behavior model, so its robustness in real complex systems has not been fully verified. In addition, the experiments are currently limited to synthetic data, which is not enough to fully explain the effect of the method in practical tasks.

**Strengths And Weaknesses:**

Strengths:
The problem that the paper focuses on is important and novel, that is, OPE is carried out under the policy dependent covariate shift caused by strategic behavior. This perspective, which is different from the standard exogenous covariate shift, is valuable. The core insight of the article is also clear: pre strategic covariates are exposed through the LID mechanism, so as to alleviate the problem of information loss under the GID setting. Methodologically, the paper forms a relatively complete technical chain from behavior modeling, drift decomposition, parameter estimation to the final construction of SDR estimator, and gives the corresponding consistent results.

Weaknesses:
The applicability of the paper depends more on the strong interactive setting of LID, but whether this setting can be obtained in many real systems is still not clear; At the same time, the method has strong structural assumptions about the behavior model, including optimization within the recommendation set, specific cost forms and conditional lognormal distribution, etc., so there are still questions about the robustness of the model mismatch. Another obvious problem is that the experiment is limited to synthetic data, and the data generation process is highly consistent with the method assumptions. Therefore, the current experiment shows that the method is more effective under the model settings, but the support for real complex scenes is still insufficient.

---

> ### Author Rebuttal · Authors · 2026-03-31
>
> We thank the reviewer for their insightful review. We are happy that the reviewer finds the problem we study is important and novel. We answer your questions below.
>
> # Regarding LID in practice and a degenerate version of our method.
> - Thank you for highlighting an area for clarification. We agree that the practicality of observing pre-strategic covariates is central to our framework.
> - LID arises in applications where decisions are made after observing an agent’s initial features and followed by personalised feedback. For example, in lending, banks observe a customer’s financial profile (e.g., income, debt, credit score), then provide actionable feedback such as reducing debt or increasing savings before reapplication. Similarly, in hiring or education platforms, applicants submit profiles, receive feedback (e.g., improve certain skills or credentials), and may subsequently update their features. In all these cases, the pre-strategic covariates are observed and stored before any strategic adaptation occurs. Such interaction patterns are already studied in strategic behavior under explanations [1,2] and algorithmic recourse [3], as well as in post-hoc explanation methods where feedback is generated for a given input [4]. These works implicitly assume access to the original input before feedback is provided. Our work adopted ARex [5] as a specific instantiation of this interaction that provides the additional structure needed for identification, while LID captures the broader and practically observed setting.
> - When $x^b$ is not observed, our framework does not admit a meaningful degenerate version. This is because the distribution of the strategic covariates becomes unidentifiable, as discussed in our Introduction and illustrated in Figure 1a. Identification then is only possible under stronger structural assumptions. For example, much of the strategic classification literature assumes that the decision maker fully knows agents’ cost functions, often further restricting them to be homogeneous across agents [6]. In contrast, our approach avoids such assumptions by leveraging LID to preserve pre-strategic information.
> - We will add these points in the revision.
>
> # Analysis on (behavioural) model mismatch.
> - Our current setting should be viewed as a simple baseline for one-shot OPE with strategic agents. We choose this specific behavioral model of agent because the DM only observes a single interaction per agent, so the available data is limited and we need a model that is easy to estimate and analyze. More flexible alternatives are possible, such as using a different positive distribution for cost sensitivity or allowing softer (non-optimal) behavioral rules, but these would typically require more data or introduce additional complexity. Our current theory is therefore a correct-specification result: if agents do not strictly optimize over the recommendation set, or if the true cost-sensitivity distribution deviates from log-normal, then the strategic transition model is misspecified and the policy value estimate may become biased. We will clarify this limitation in the revision.
> - At the same time, the assumption that agents optimize within the recommendation set is reasonable in our setting. An agent changes their covariates from $x^b$ to $x^s$ only when this choice maximizes their utility. If a change is costly or infeasible, this is reflected in the cost term of the utility. Moreover, as mentioned in L120-L125 (right column) we assume agents do not explore options outside the recommendation set, as doing so may risk reducing their utility.
>
> # Additional experiments and stronger stress tests.
> - Thank you for your insightful comment. We agree that additional experiments with real-world data and stronger stress tests would strengthen our results more; and we will add these in the revision.
> - At the same time, we want to re-emphasise that while our contributions in terms of estimation are still primarily theoretical, the contributions in terms of problem setting are novel and realistic, which brings attention to the issue of information asymmetry and how specific interaction schemes can enable learning hidden agents’ behaviour.
>
> References:
> - [1] Tsirtsis, Stratis, and Manuel Gomez Rodriguez. "Decisions, counterfactual explanations and strategic behavior." NeurIPS (2020).
> - [2] Xie, Tian, and Xueru Zhang. "Non-linear welfare-aware strategic learning." AAAI (2024).
> - [3] Karimi, Amir-Hossein, et al. "A survey of algorithmic recourse: definitions, formulations, solutions, and prospects." arXiv preprint arXiv:2010.04050 (2020).
> - [4] Lundberg, Scott M., and Su-In Lee. "A unified approach to interpreting model predictions." NeurIPS (2017).
> - [5] Vo, Kiet QH, et al. "Explanation Design in Strategic Learning: Sufficient Explanations that Induce Non-harmful Responses." AISTATS (2026).
> - [6] Rosenfeld, Elan, and Nir Rosenfeld. "One-Shot Strategic Classification Under Unknown Costs." ICML (2024).

---

> > ### Author Rebuttal · Reviewer_V8eb · 2026-04-03
> >
> > Thank you for the rebuttal. The response clarifies the intended setting of LID and helps explain why the method does not have a natural degenerate version without observing pre-strategic covariates. However, my main concerns are only partially resolved. In particular, the practicality of the LID setting in real applications is still not fully convincing, the discussion of behavioral model misspecification remains mostly qualitative, and the paper still lacks stronger stress tests or experiments beyond synthetic settings. Overall, the rebuttal improves the clarity of the paper, but I still have reservations about the practical applicability and robustness of the method.

---

> > > ### Author Response · Authors · 2026-04-08
> > >
> > > Thank you for your thoughtful feedback. To address the reviewer’s remaining concerns, we have conducted additional experiments to demonstrate the practicality and robustness of our approach. We report the results below.
> > >
> > > In the synthetic setup, we let a portion of agents behave **irrationally** by choosing a feature update arbitrarily from the space of feature values. The table below shows the median of the errors $\hat{V}_{SDR}-V$ of the noisy SDR estimates. As the dataset size increases, the variance of those estimates also reduces.
> > >
> > > | N | 2000 | 6000 | 10000 |
> > > |---|---|---|---|
> > > | $p_{irr}=0.00$ | -0.01 | 0.02 | -0.01 |
> > > | $p_{irr}=0.01$ | 0.40 | 0.42 | 0.45 |
> > > | $p_{irr}=0.02$ | 0.51 | 0.50 | 0.49 |
> > >
> > > This shows that when agents behave irrationally, the SDR estimator can be biased, as we cannot anticipate the strategic covariate shift correctly. In addition, as the rate of irrational agents, $p_{irr}$, is kept fixed, increasing the dataset size does not help. From the table, as the amount of bias decreases when the rate $p_{irr}$ decreases, it suggests that if we keep the number of irrational agents fixed while increasing the size of the dataset, the bias should approach zero. We plan to add this additional stress test to the revision.
> > >
> > > In addition, we also provide the result of an experiment with the **German credit dataset** to demonstrate the practicality of LID and our SDR estimator. We refer to our response to Reviewer ``wSgu`` for the result.
> > >
> > > **Our action plan:** We will finalise the results on the stress tests of our model and German Credit dataset and include them in the camera-ready version of the main paper. We believe these results will more clearly demonstrate the practical applicability and robustness of the proposed method. We thank the reviewer for helping us strengthen this aspect of the paper.

---

### Official Review · Reviewer_ixLY · 2026-03-10

**Soundness:** 3
**Presentation:** 3
**Significance:** 3
**Originality:** 3
**Overall Recommendation:** 4
**Confidence:** 3

**Summary:**

The paper considers the challenge of off-policy evaluation (OPE) under strategic behavior that shifts the covariates, i.e., policy-dependent covariates. They consider the local information disclosure (LID) design, in which agents first submit their original covariates. The decision maker then provides feedback, after which the agents submit their modified covariates. Under this problem setting and certain assumptions, they introduce a strategy-robust DR estimator by learning the agents' cost model and strategic covariates shift.

**Compliance With Llm Reviewing Policy:**

Affirmed.

**Final Justification:**

I’ve read the rebuttal and will maintain my current acceptance score.

**Key Questions For Authors:**

Maybe I'm misunderstanding this point. The paper models the outcome that depends on the final covariate x^s but is independent of the initial covariate x^b. This seems inconsistent with the bank example, where agents can shift their credit-related covariates, and the final outcome is entirely independent of the initial profile. I would think there is a latent variable for the agents' intrinsic default risk that does not change much. Can the author further justify this outcome model choice?

**Strengths And Weaknesses:**

The paper is well written and focuses on an interesting strategic setting in OPE. Here are some of my comments:

1. The strategic agent problem in OPE is well motivated, and I understand that the GID and LID are design choices, i.e., part of the problem formulation but not the solution approach. It would be helpful if the authors could better motivate this particular LID choice.
2. The assumption of heterogeneous agents with the log-normal covariate shift sensitivity and a mean linear in the covariates seems to be specific. Can the authors provide further justification?
3. The finite covariate space X is a major limitation. Is the extension to a continuous space straightforward in this setting?

---

> ### Author Rebuttal · Authors · 2026-03-31
>
> We thank the reviewer for their encouraging and insightful review. We are happy that the reviewer finds our paper well written and has an interesting strategic setting in OPE. We address the reviewer concerns below.
>
> # Motivation for this particular LID choice.
> - The LID interaction scheme offers several advantages to both the decision maker (DM) and the agents, as opposed to the GID interaction scheme. In contrast to the GID setting where the DM discloses the same information to every agent, the DM in LID gives personalised feedback to agents. Such personalised information can be more relevant and helpful to a specific agent for improving their qualification. For instance, if a bank only discloses feature importance weights of their decision making model to all agents, these are not really helpful at telling agents how they should modify their features to obtain the desired outcome. This corresponds to a GID setting. In contrast, if a bank queries pre-strategic information from agents, they can give specific recommendations to each of those agents about how much they should change their features and what they can expect at the end.
> - Although our Introduction briefly mentioned many works that have examined both GID and LID settings, we will elaborate these points in the revision.
>
> # Justification for the specific log-normal model.
> - Our choice of log-normal model has a natural motivation based on the central limit theorem (CLT): $\alpha>0$ is a positive scale parameter that describes an agent’s overall cost sensitivity, which can be interpreted as the aggregate effect of many latent factors (e.g., liquidity, time constraints, opportunity cost, motivation, risk tolerance etc.). If these factors combine multiplicatively on the original scale, they combine additively on the log scale, so that $\ln(\alpha)\mid x^b$ is the sum of many latent contributions (that do not need to be normally distributed). If their distributions have finite variance, the standard CLT allows approximating $\ln(\alpha)\mid x^b$ (as their sum) by a normal distribution. We will add this motivation and also state more clearly that log-normality is a CLT-justified modeling choice rather than a claim of exact behavioral truth. Note that this kind of assumption is the backbone of Gaussian or Student-$t$-tests throughout empirical science, where data are never exactly normally distributed.
> - We model the mean parameter as a linear function in the embeddings of covariates for interpretability and to keep the number of parameters small. Since the DM does not rely on multiple interactions with agents, we must estimate parameters from only one interaction per agent (one-shot), which implies limited data. The transformation $\phi(x^b)$ allows non-linear relationships between the mean (of the log-normal distribution) and covariates. This choice also makes identification and estimation of the parameters easier to analyze. We will clarify this point in the revision.
>
> # Extension to a continuous covariate space.
> While our current analysis is presented for a discrete covariate space, the main idea can be extended to a continuous covariate space under suitable additional regularity conditions. In particular, the theory would require modifications to the proof arguments, such as replacing finite-sum expressions with integral-based formulations, density-based overlap assumption, and imposing appropriate compactness, continuity to ensure identifiability and uniform convergence. We leave this extension for future work.
>
> # Regarding the outcome $Y$ not directly depending on $X^b$.
> - We agree that in many real-world applications, such as lending, agents may possess latent characteristics (e.g., intrinsic default risk) that are not fully captured by observable covariates and may persist despite strategic modifications.
> - Our modeling choice can be interpreted as a sufficiency assumption: we assume that the post-adaptation covariates $x^s$ , together with the treatment $t^s$, contain all outcome-relevant information observable to the decision maker. Any remaining unobserved factors are captured by the noise variable $z$, which we assume to be exogenous (Lines 175-177). This does not imply that agents cannot **game** the system or that (endogenous) intrinsic characteristics do not exist. Rather, our framework focuses on settings where feature changes correspond to meaningful improvements (e.g., reducing debt, increasing savings), so that the updated covariates $x^s$ are informative of the agent’s actual risk profile. In such settings, conditioning on $x^s$ is a reasonable approximation for outcome modeling.
> - We acknowledge that if there exist persistent latent factors affecting the outcome that are not captured by $x^s$, then the conditional independence assumption may fail. Although we discussed a challenge posed by this extension in our Conclusion, we are happy to elaborate this point in the revision.

---

> > ### Author Rebuttal · Reviewer_ixLY · 2026-04-03
> >
> > I would like to thank the authors for their detailed responses. At this stage, I am inclined to retain my current acceptance score.

---

> > > ### Author Response · Authors · 2026-04-08
> > >
> > > Thank you for your response. To supplement our previous rebuttal, we would like to refer to our response to Reviewer ``wSgu`` for our additional experimental result on the German credit dataset. We believe this experiment further clarifies your concerns as it demonstrates (1) how this particular LID choice can be implemented in a practical scenario and (2) how our approach can be adapted to handle continuous features.

---

### Official Review · Reviewer_wSgu · 2026-03-12

**Soundness:** 2
**Presentation:** 3
**Significance:** 2
**Originality:** 2
**Overall Recommendation:** 3
**Confidence:** 4

**Summary:**

This paper studies off-policy evaluation in environments where agents may respond strategically to the decision rule. The motivation is that standard OPE methods assume a fixed covariate distribution, while strategic behavior can induce policy-dependent covariate shifts, potentially invalidating classical evaluation approaches.
To address this, the paper combines two currently active research directions, off-policy evaluation and strategic classification, and investigates how to evaluate a target policy when agents can manipulate their observed features. The authors focus on a local information disclosure setting in which pre-strategic covariates are observable, propose a model of strategic responses based on a recommendation set, and develop a strategy-robust doubly robust estimator with identification and consistency guarantees

**Compliance With Llm Reviewing Policy:**

Affirmed.

**Final Justification:**

The authors have addressed most of my concerns.

However, the overall theoretical contribution is somewhat limited, and the experimental evaluation remains insufficient. I think this work is hardly meeting the level expected for ICML.

I also acknowledge the improvements and the authors’ efforts during the rebuttal. Based on these, I am willing to raise my score to 3.

**Key Questions For Authors:**

Please see the weaknesses above.

**Limitations:**

yes

**Strengths And Weaknesses:**

**Strengths:**

- The work considers off-policy evaluation (OPE) in environments where agents may respond strategically to the decision rule. Since standard OPE typically assumes a fixed covariate distribution, examining how strategic behavior may invalidate this assumption raises a meaningful and timely research question.

- The authors introduce a local information disclosure (LID) setting in which pre-strategic covariates can be observed, and combine it with a recommendation-set-based interaction protocol to model agents’ strategic responses. This provides a coherent modeling framework for studying the interaction between strategic behavior and policy evaluation.

- Under the specified structural assumptions, the paper establishes identification results for the behavioral model and derives consistency guarantees for the proposed strategy-robust estimator. These results provide an initial theoretical step toward understanding OPE in strategic environments.

**Weaknesses**

- The main theoretical component of the paper builds on the standard strategic classification framework, where agents follow a best-response model under a cost function. The proposed formulation mainly adapts this framework to the OPE setting by introducing a recommendation set that discretizes the agents’ action space. As a result, the contribution appears largely combinational, integrating existing strategic classification modeling with the OPE setting, without introducing sufficiently novel theoretical insights. Moreover, discretizing the agents’ action space simplifies the best-response problem and reduces the complexity of strategic interactions compared to the more general settings typically studied in the strategic classification literature. Finally, the theoretical guarantees rely on several strong structural assumptions, which may be difficult to satisfy in realistic strategic environments.

- The paper’s treatment of strategic behavior appears somewhat limited in terms of engagement with recent literature that studies how agents actually respond in strategic environments. For example, the recent work “The Double-Edged Sword of Behavioral Responses in Strategic Classification: Theory and User Studies” (FAccT 2025) highlights that real human responses to algorithmic decision rules can deviate substantially from the idealized best-response models commonly assumed in strategic classification. In such settings, agents’ behavioral choices may not align with the simplified decision structure assumed by the system. Consequently, when the proposed framework discretizes the action space through a recommendation set, individuals’ actual responses may not necessarily follow the options provided by the recommendation set. This raises potential concerns about the behavioral realism of the proposed modeling framework and suggests that this limitation should be discussed more explicitly.

- The experimental evaluation is relatively limited and is conducted primarily on synthetic data. The experiments mainly serve as sanity checks for the theoretical results rather than providing strong empirical validation of the proposed method. In both the OPE literature and the broader strategic machine learning literature, it is common to include experiments on real or semi-synthetic datasets to demonstrate practical relevance. The absence of such evaluations makes it difficult to assess the practical impact of the proposed approach.

---

> ### Author Rebuttal · Authors · 2026-03-31
>
> Thank you for your insightful feedback. We are happy that the reviewer appreciates our work. We address your concerns below.
>
> # On novel theoretical insights
> - We respectfully disagree with the reviewer that our work is largely combinational without sufficient novel theoretical insights. As mentioned in our Introduction and Conclusion, our work (1) brings to attention the challenges of information asymmetry which is often overlooked in strategic classification literature, (2) shows that this asymmetry issue can be mitigated **realistically** through designing appropriate interaction schemes, and (3) demonstrates that such LID schemes can enable one-shot learning under unknown and heterogeneous agents behaviour. We connect these insights to our technical contributions and prior work below.
> - Although we adopt ingredients from prior work in strategic machine learning, to our knowledge, **it has not been observed that pre-strategic information can be leveraged to predict strategic covariate shift, especially when agents’ behavioural model is heterogeneous and not fully known to the decision maker (DM)**. For instance, [1] points out that prior work in *offline* strategic classification relies on fully known agents’ behavioural models. Although their work is a pioneer in learning under one-shot settings without full knowledge of agents’ model, they do not leverage pre-strategic information and therefore assume all agents have the same cost function.
> - On the other hand, prior work in causal effect estimation and policy evaluation/learning still relies on repeated deployment of policies. **There is no work that has observed the advantage of using pre-strategic information for inferring agents’ behavioural model**.
> - Although relying on strong structural assumptions, our work is **the first to demonstrate the advantage of pre-strategic information in offline learning settings**, with unknown and heterogeneous agents’ behaviour. We view these structural assumptions as the necessary first step towards enabling one-shot learning under unknown and heterogeneous behaviour, and leave extensions to future work.
> - We will make these points clearer in the revision.
>
> # On agents' behavioural model
> - We agree with the reviewer that real-world agents may not behave as assumed by the system, as is **always the case in any algorithmic systems that require a model of human behaviour**. We clarify your concern in detail below.
> - The agents’ behavioural model is reasonable in our setting. As pointed out by the authors of ARexes [3], in many real-world scenarios, the DM can conduct surveys to better generate recommendations that an agent prefers, reducing the chance of an agent deviating from the recommendations. In our work, we extended the ARex framework further to allow for an unrestricted number of recommendations, which further reduces the chance of an agent deviating from the recommendations.
> - Our agents’ behavioral model does not contradict recent results in strategic classification literature. While the recent work by Ebrahimi et al. (2025) highlights an interesting point that human responses can deviate from the commonly assumed best response models, the work operates in a different setting, where the agents are provided feature importance weights as explanations and they are on their own for interpreting how to put together the weights to anticipate the benefit of a potential action. As pointed out in the work of Vo et al. (2026), such uncertainty (which is common in many explanation types) gives rise to misinterpretation. In contrast, our work adopts the ARex framework, which is transparent to the agents about the benefit of recommended actions, and can avoid the agents’ act of guessing the benefit. This behavioural model is also consistent with the work of [4].
> - We will discuss this more in the revision.
>
> # Additional experiments
> - We agree that additional experiments with real-world data would strengthen our results more; and we will add these in the revision.
> - At the same time, we want to re-emphasise that while our contributions in terms of estimation are still primarily theoretical, the contributions in terms of problem setting are novel and realistic, which brings attention to the issue of information asymmetry and how specific interaction schemes can enable learning hidden agents’ behaviour.
>
> References:
> - [1] Rosenfeld, Elan, and Nir Rosenfeld. "One-Shot Strategic Classification Under Unknown Costs." ICML (2024).
> - [2] Ebrahimi, Raman, Kristen Vaccaro, and Parinaz Naghizadeh. "The double-edged sword of behavioral responses in strategic classification: Theory and user studies." FAccT (2025).
> - [3] Vo, Kiet QH, et al. "Explanation Design in Strategic Learning: Sufficient Explanations that Induce Non-harmful Responses." AISTATS (2026).
> - [4] Tsirtsis, Stratis, and Manuel Gomez Rodriguez. "Decisions, counterfactual explanations and strategic behavior." NeurIPS (2020).

---

> > ### Author Rebuttal · Reviewer_wSgu · 2026-04-04
> >
> > In their rebuttal, the authors emphasize the theoretical contributions of the paper, which addresses part of my concern. However, my concerns regarding the experimental evaluation are largely not addressed. The rebuttal does not provide additional experimental results, nor does it sufficiently clarify the experimental setup. This makes it difficult for me to increase my score at this stage.

---

> > > ### Author Response · Authors · 2026-04-08
> > >
> > > Thank you for your thoughtful feedback. We are glad that our rebuttal addressed parts of your concerns related to the theory. To address the reviewer’s remaining concerns regarding the experimental evaluation, we have conducted an additional experiment with the German credit data [1] in order to demonstrate the practicality of the proposed approach. The results are reported below.
> > >
> > > - The German credit dataset consists of 18 features of mixed types (categorical & numerical values) after removing sensitive features. Additional data are generated using CTGAN [2] fitted on the original dataset.
> > > - We then train a logistic regression model and use it as the base model for constructing the logging policy and evaluation policy, by scaling the learned coefficients by different constant values.
> > > - To operationalise ARexes, we let the DM generate all possible candidates for feature updates, and for each agent, the DM gives top 3 recommendations $x^r$ with highest value of $\pi(x^r)/d(x^r,x^b)$. Such recommendations aim to balance the benefit and the (base) effort necessary for an agent to adapt. For example, a bank might recommend a change in a customer's profile such that it can benefit the customer without incurring high cost for them.
> > > - As the features are now continuous, we use Monte Carlo approximation to compute the SDR (policy-value) estimate. In particular, we do not fit a model for $p(x^b)$, we instead compute only $p(t^s,x^s,t^b|x^b)$ and use it to weigh the data points when computing the SDR estimate.
> > >
> > > The **ground-truth policy value is 16.73**, our **SDR estimator gives 16.77**, and the **S-IPS estimator gives 15.46** (likely due to numerical instability when evaluating the importance weights). We report the result below (the median errors of noisy estimates) with different dataset sizes. From the plots, we observe that there is a trend of convergence (as more noisy estimates approach the ground-truth as the dataset size increases).
> > >
> > > | N | 9000 | 50000 | 90000 |
> > > |---|---|---|---|
> > > | $\|\hat{\theta}-\theta*\|_2$ | 1.18 | 0.89 | 0.88 |
> > > | $\hat{V}_{SDR}-V$ | 0.0124 | 0.0123| 0.0124 |
> > >
> > > Because the parameter vector $\theta$ of the log-normal distribution has 20 dimensions (which were 4 in the previous synthetic setup), this introduces difficulty in learning these parameters, hence the slow convergence.
> > >
> > > In addition, we provide the result of a robustness study when some agents do not act as we assume. We refer to our response to Reviewer ``V8eb`` for the result.
> > >
> > > **Our action plan:** We will finalise the results on the German Credit dataset and the stress tests of our behavioural model; and include them in the camera-ready version of the main paper. We believe these results will more clearly demonstrate the practical relevance and impact of our work. We thank the reviewer for helping us strengthen this aspect of the paper.
> > >
> > > References:
> > > - [1] Hofmann, Hans. "Statlog (German credit data)." UCI Machine Learning Repository 10 (1994): C5NC77.
> > > - [2] Xu, Lei, et al. "Modeling tabular data using conditional gan." Advances in neural information processing systems 32 (2019).

---

### Official Review · Reviewer_Yebn · 2026-03-20

**Soundness:** 4
**Presentation:** 4
**Significance:** 3
**Originality:** 3
**Overall Recommendation:** 5
**Confidence:** 4

**Summary:**

In this paper, the authors study Off-Policy Evaluation (OPE) in a setting where
the agents will strategically alter their features (covariates), after the
policy is changed by the decision maker (DM). The difficulty in this setting,
is that information about the agent can be lost, when they change the
covariates. The authors introduce an assumption, called Local Information
Disclosure, that allows them to ensure that enough information is observed by
the DM. The authors adopt a recently introduced algorithm (ARex, [1]) and show
that this would supply the LID requirement. Extending on this observation, the
authors introduce altered versions of standard OPE estimators for their
setting. They introduce a strategic version of the inverse propensity score
(IPS) and Direct Method (DM), which can be combined for a strategic version of
the Doubly Robust. Proofs of the consistency of these estimators are also
provided.

The authors provide experiments on synthetic data to verify their theoretical
results. The results show that their estimator is able to learn the true
parameters even in this difficult strategic environment, while the standard
estimators fail in such a setting.

[1] Vo, K. Q., Chau, S. L., Kato, M., Wang, Y., and Muandet, K. Explanation
design in strategic learning: Sufficient explanations that induce non-harmful
responses.

**Compliance With Llm Reviewing Policy:**

Affirmed.

**Final Justification:**

I have read the responses in the rebuttal and remain my score.

**Key Questions For Authors:**

1. The authors claim in the introduction that they consider a setting where
   agents can have different preferences. However, in their model the only
   difference between agents is the weight each agent puts on the effort to
   change the features and the effort is the same for each agent. Could the
   authors adapt their framework to a setting where each agent can have
   different cost functions? For example, one agent would fine feature 1 easy
   to alter, but feature 2 difficult, while there is another agent that finds
   feature 1 difficult the alter and feature 2 easy.
2. It is assumed that the covariate data is discrete, and the authors argue that
   this is realistic in most cases. However, I would still be interested to
   know if they think their proofs and procedures can be extended to a
   continuous domain. Maybe with some restrictions like, convex and compact.
3. The policy in this setting only gives the probability of a binary treatment,
   could your strategic estimators be extended to a setting where to policy
   gives a distribution over multiple possible actions?
4. How is the LID setting different from the setting commonly assumed in the
   counterfactual explanation literature? If it is any different. The procedure
   of showing your features, getting a prediction and personalized feedback, and
   then changing the features seems very similar to the setting commonly
   discussed in counterfactual explanation papers.

**Limitations:**

yes

**Strengths And Weaknesses:**

A clear strength is that this setting has not been researched before and is
related to the developments in strategic classification and performative
prediction. Studying OPE in a strategic setting is thus a natural and
interesting combination of these fields. The writing is clear and the set up is
written in an understandable concise manner. I appreciate the inclusion of the
proofs of their claims, which are good to follow and look to be correct. The
experiments strengthen the claims and show that the proposed estimator works in
practise as well.

Some of the results do hinge on the assumption on $\alpha$ being log-normally
distributed, which seems to be a natural assumption. Especially, because it
allows for explicit expressions in the proofs. However, it seems restrictive
and it would have been appreciated if some discussion on possible extensions
was added, or reasons why it is less restrictive. On top of that, the authors
claim that the $\alpha$ allows to model agent preference, but it does not allow
agents to prefer one feature change over another, only the overall strength of
the cost function.

---

> ### Author Rebuttal · Authors · 2026-03-31
>
> We thank the reviewer for the encouraging and insightful review. We address each of your concerns below.
>
> # On $\alpha$ being log-normally distributed.
> - We refer the reviewer to our answer to question 2 of Reviewer ixLY, who also raised a similar point. We provided a detailed response that our modelling choice is based on the central limit theorem. We will expand our justification in the revision.
>
> # Question 1: Extending the framework to allow agents with diverse cost function forms.
> - The problem formulation can be extended to accommodate the case where agents have different cost function forms. For example, if different agents prefer to change different features, this can be modelled through a vector of cost sensitivity rather than just a scalar. In particular, one can define the cost as $c(x,x^\prime) = \alpha^\top d(x,x^\prime)$. However, this leads to complications in analysing the strategic covariate shift $p(x^s|t^b,x^b)$, as this would involve deriving the distribution of the random variable $\alpha^\top d(x,x^\prime)$ from the distribution of the random vector $\alpha$.
>
> # Question 2: Extending the theory to continuous domains.
> - As discussed in Lines 120–124, the discrete covariate space covers many realistic settings, especially when covariates are naturally categorical or when discretization is beneficial for small datasets. This discretization can also facilitate the overlap/positivity condition in Assumption 3.2, because finite support may reduce the prevalence of very small propensity scores and thereby stabilize the importance weights used in the SDR estimator (Section 3).
> - While our current analysis is presented for a discrete covariate space (mainly to prove the consistency result), the main idea can be extended to a continuous covariate space under suitable *additional regularity conditions*. In particular, the theory would require modifications to the proof arguments, such as replacing finite-sum expressions with integral-based formulations, density-based overlap assumption, and imposing appropriate compactness, continuity to ensure identifiability and uniform convergence. We leave this extension for future work.
>
> # Question 3: Extending the estimators to non-binary treatment cases.
> - Our strategic estimators can be straightforwardly extended to the case of non-binary treatment (i.e., with 3 or more treatment values). The only challenge lies in the DM-agent interaction of LID, as it requires a different approach to model the potential benefit the agent receives from a particular treatment value (which previously was just $\pi(x)=p(T=1|x)$). This is because in non-binary treatment cases, it is unclear which specific treatment option an agent might prefer. Besides this, the remainder of the estimator derivation does not require the treatment variable to be binary.
>
> # Question 4: LID vs. counterfactual explanations.
> - The ARex framework (which we used as a LID setting), adopted from the work of Vo et al. (2026), indeed has a strong connection to the counterfactual explanation literature. Two major distinctions motivate our use of this framework.
> - Firstly, the counterfactual explanation literature focuses more on the **epistemic** aspect of explanations (i.e., providing understanding) and algorithms to generate the recommended actions rather than on the **strategic** aspect of how agents react when receiving such recommendations. The ARex framework is positioned in strategic settings and therefore comes with a response model for agents, which we can use in our OPE problem.
> - Secondly, in contrast to the common goal of prior work in counterfactual explanations as providing **minimal changes** (such that prediction scores can improve), we do not have such a restriction and specificity that an explanation algorithm must follow. Rather, we only require that the recommendations have different distances to $x^b$ (i.e., in Lemma 2.2.) and any simple algorithm for generating such recommendations suffices.

---

> > ### Author Rebuttal · Reviewer_Yebn · 2026-04-03
> >
> > I thank the reviewer for their answers and retain my score.
> >
> > One response to the answer about modelling diverse cost functions I would like to add, is that the notation $\alpha^{\top} d(x, x')$ is slightly confusing, as $d \colon \mathcal{X} \times \mathcal{X} \to \mathbb{R}$, but I assume that the authors mean that the weight gets applied to to the features, or that the distance function can be decomposed to individual contributiosn. Intuitively, with the Euclidean norm you could apply a PSD matrix $(x - x')^{\top} \Lambda (x -x')$, and get ellipsoids pointing in different directions. I understand this complicates the analysis, but it would be interesting for in the future!

---

> > > ### Author Response · Authors · 2026-04-08
> > >
> > > Thank you for your insightful feedback and suggestion. We agree that such general cost functions would be an interesting future direction.
> > >
> > > Additionally, we would like to refer to our response to Reviewer ``wSgu`` where we provide an additional experimental result with the German credit dataset. This experiment demonstrates how our approach can be adapted to handle continuous features.

---

### Decision · Program_Chairs · 2026-04-30

**Decision:**

Accept (regular)

**Comment:**

The paper studies a novel and interesting problem, and on balance I think it clears the bar. The reviewers agreed that the formulation is clean and the technical development is solid, with a coherent path from behavioral modeling to identification and estimation. The main reservations concern the strength and realism of the LID and strategic-response assumptions, limited empirical validation beyond synthetic settings, and the need for clearer positioning relative to strategic classification and related work. The rebuttal improved clarity but did not fully resolve these points, so I ask the authors to discuss these limitations and scope more explicitly in the final version.